# Neural Relation Graph: A Unified Framework for Identifying Label Noise and Outlier Data

**Jang-Hyun Kim**
Seoul National University
janghyun@mllab.snu.ac.kr

**Sangdoo Yun**
NAVER AI Lab
sangdoo.yun@navercorp.com

**Hyun Oh Song**[*]
Seoul National University
hyunoh@mllab.snu.ac.kr

## Abstract

Diagnosing and cleaning data is a crucial step for building robust machine learning systems. However, identifying problems within large-scale datasets with real-world distributions is challenging due to the presence of complex issues such as label errors, under-representation, and outliers. In this paper, we propose a unified approach for identifying the problematic data by utilizing a largely ignored source of information: a relational structure of data in the feature-embedded space. To this end, we present scalable and effective algorithms for detecting label errors and outlier data based on the relational graph structure of data. We further introduce a visualization tool that provides contextual information of a data point in the feature-embedded space, serving as an effective tool for interactively diagnosing data. We evaluate the label error and outlier/out-of-distribution (OOD) detection performances of our approach on the large-scale image, speech, and language domain tasks, including ImageNet, ESC-50, and SST2. Our approach achieves state-of-the-art detection performance on all tasks considered and demonstrates its effectiveness in debugging large-scale real-world datasets across various domains. We release codes at `https://github.com/snu-mllab/Neural-Relation-Graph`.

## 1 Introduction

Identifying problems within datasets is crucial for improving the robustness of machine learning systems and analyzing the model failures [45]. For instance, identifying mislabeled or uninformative data helps construct concise and effective training datasets [33], while identifying whether test data is OOD or corrupted allows for more accurate model evaluation and analysis [54].

In recent years, efforts have been made to identify problematic data by utilizing unary scores on individual data from trained models, such as estimating data influence [22], monitoring prediction variability throughout training [52], and calculating prediction error margins [34]. However, identifying such data can be challenging, particularly when dealing with large-scale datasets from real-world distributions. In real-world settings, datasets may have complex problems, including label errors, under-representation, and outliers, each of which can lead to the model error and prediction sensitivity [23]. For example, Figure 1 shows that a neural network exhibits low negative prediction margins and high loss values for both a sample with label error and outlier data. This observation indicates that previous unary scoring methods may have limitations in discerning whether the problem lies with the label or the data itself.

In this work, we propose a unified framework for identifying label errors and outliers by leveraging the feature-embedded structure of a dataset that provides richer information than individual data alone [47, 36]. We measure the relationship among data in the feature embedding space while comparing the assigned labels independently. By comparing input data and labels separately, we are able to

---

[*]Corresponding author

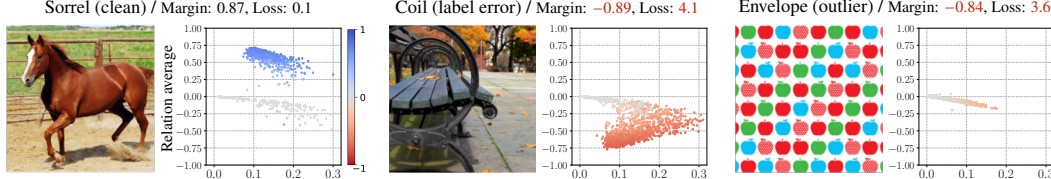

Figure 1: ImageNet samples with their labels and the corresponding relation maps by an MAE-Large model [13]. We report the prediction margin score ($\in [-1, 1]$) and the loss value next to the label. The relation map draws a scatter plot of the mean and variance of relation values of a data pair throughout the training process. Here the color represents the relation value at the last converged checkpoint. We present the detailed procedure for generating the relation maps in Section 3.5.

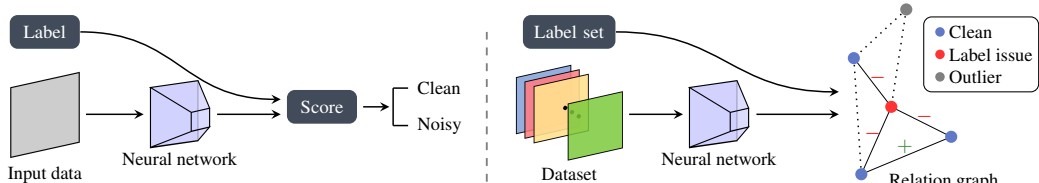

Figure 2: The conceptual illustration of the conventional approaches (left) and our proposed approach (right). In the relation graph, positive edges signify complementary relations, negative edges denote conflicting relations, and dashed lines indicate negligible relations between data.

isolate the factors contributing to model errors, resulting in improved identification of label errors and outlier data, respectively. Based on the relational information, we construct a novel graph structure on the dataset and identify whether the data itself or the label is problematic (Figure 2). To this end, we develop scalable graph algorithms that accurately identify label errors and outlier data points.

In Section 3.5, we further introduce a visualization tool named data relation map that captures the relational structure of a data point. Through the map, we can understand the underlying relational structure and interactively diagnose data. In Figure 1, we observe different patterns in the relation maps of the second and third samples, despite their similar margin and loss scores. This highlights that the relational structure provides complementary information not captured by the unary scoring methods.

Our approach only requires the model's feature embedding and prediction score on data, making it more scalable compared to methods that require calculating the network gradient on each data point or retraining models multiple times to estimate data influence [39, 17]. Furthermore, our method is domain- and model-agnostic, and thus is applicable to various tasks. We evaluate our approach on label error and outlier/OOD detection tasks with large-scale image, speech, and language datasets: ImageNet [43], ESC-50 [37], and SST2 [55]. Our experiments show state-of-the-art performance on all tasks, demonstrating its effectiveness for debugging and cleaning datasets over various domains.

## 2   Related works

**Label error detection**   Label errors in datasets can negatively impact model generalization and destabilize evaluation systems [16, 34]. Prior works address this issue through label error detection using bagging and bootstrapping [46, 41], or employing neural networks [18, 27, 9, 19]. To mitigate overfitting on label errors, Pleiss et al. [38] propose tracking the training process to measure the area under the margin curve. Recent studies demonstrate that simple scoring methods with large pre-trained models, such as prediction margins or loss values, achieve comparable results to previous complex approaches [33, 5]. Meanwhile, Wu et al. [57] propose a unified approach for learning with open-world noisy data. However, the method involves a complicated optimization process during training, which is not suitable for large-scale settings. Another line of approach to identifying label errors involves measuring the influence of a training data point on its own loss [22, 39]. However, these approaches require calculating computationally expensive network gradients on each data point, and their performance is known to be sensitive to outliers and training schemes [2, 3]. In this work, we present a scalable approach that leverages the data relational structure of trained models without additional training procedures, facilitating practical analysis of label issues.

**Outlier/OOD detection**   Detecting outlier data is crucial for building robust machine learning systems in real-world environments [23]. A recent survey paper defines the problem of finding outliers in training set as *outlier detection* and finding outliers in the inference process as *OOD detection* [59]. The conventional approach for detecting outliers involves measuring k-nearest distance using efficient sampling methods [48]. More recently, attempts have been made to detect outlier data using scores obtained from trained neural networks, such as Maximum Softmax Probability [14], Energy score [29], and Max Logit score [15]. Other approaches suggest adding perturbations on the inputs or rectifying the activation values to identify the outlier data [28, 49]. Lee et al. [26] propose fitting a Gaussian probabilistic model to estimate the data distribution. Recently, Sun et al. [50] propose a non-parametric approach measuring the $k$-nearest feature distance. In our work, we explore the use of the relational structure on the feature-embedded space for identifying outlier data. Our approach is applicable to a wide range of domains without requiring additional training while outperforming existing scoring methods on large-scale outlier/OOD detection benchmarks.

## 3   Methods

In this section, we describe our method for identifying label errors and outliers using a model trained on the noisy training dataset. We exploit the feature-embedded structure of the learned neural networks, which are known to effectively capture the underlying semantics of the data [40]. We define data relation to construct a data relation graph on the feature space, and introduce our novel graph algorithms for identifying label errors and outlier data. In Section 3.5, we introduce the data relation map as an effective visualization tool for diagnosing and contextualizing data.

### 3.1   Data relation

We describe our approach in the context of a classification task, while also noting that the ideas are generalizable to other types of tasks as well. We assume the presence of a trained neural network on a noisy training dataset with label errors and outliers, $\mathcal{T} = \{(x_i, y_i) \mid i = 1, \ldots, n\}$. By utilizing data features extracted from the network, we measure the semantic similarity between data points with a bounded kernel $k : \mathcal{X} \times \mathcal{X} \to [0, M]$, where a higher kernel value indicates greater similarity between data points. Our framework can accommodate various bounded kernels such as RBF kernel or cosine similarity [60]. We provide detailed information on the kernel function used in our main experiments in Section 3.4.

By incorporating the assigned label information with the similarity kernel $k$, we define the relation function $r : \mathcal{X} \times \mathcal{Y} \times \mathcal{X} \times \mathcal{Y} \to [-M, M]$:

$$r\left((x_i, y_i), (x_j, y_j)\right) = \mathbb{1}(y_i = y_j) \cdot k(x_i, x_j), \quad (1)$$

where $\mathbb{1}(y_i = y_j) \in \{-1, 1\}$ is a signed indicator value. The relation function reflects the degree to which data samples are complementary or conflicting with each other. In Figure 3, the center image with a label error has negative relations to the left samples that belong to the same ground-truth class. In contrast, the two left samples with correct labels have a positive relation. We also note that samples with dissimilar semantics exhibit near-zero relations.

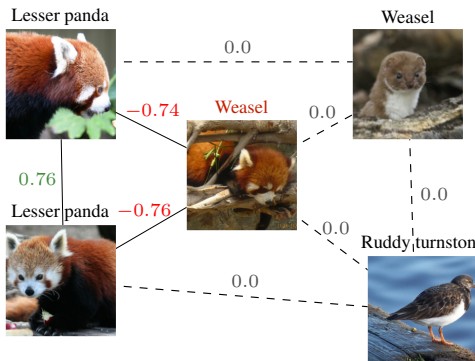

Figure 3: Relation values of samples from ImageNet with MAE-Large [13]. We denote the assigned label above each sample. Here, the center image has a label error.

Our relation function $r$ relies solely on the parallelizable forward computation of neural networks, ensuring scalability in large-scale settings.

### 3.2   Label error detection

We consider a fully-connected undirected graph $\mathcal{G} = (\mathcal{V}, \mathcal{E}, \mathcal{W})$, where the set of nodes $\mathcal{V}$ corresponds to $\mathcal{T}$ and the weights $\mathcal{W}$ on edges $\mathcal{E}$ are the negative relation values defined in Equation (1). For notation clarity, we denote a data point by an index, *i.e.*, $\mathcal{T} = \{1, \ldots, n\}$. Then, for nodes $i$ and $j$, the edge weight is $w(i, j) = -r(i, j) = -r((x_i, y_i), (x_j, y_j))$. We set $w(i, i)$ to 0, which does not

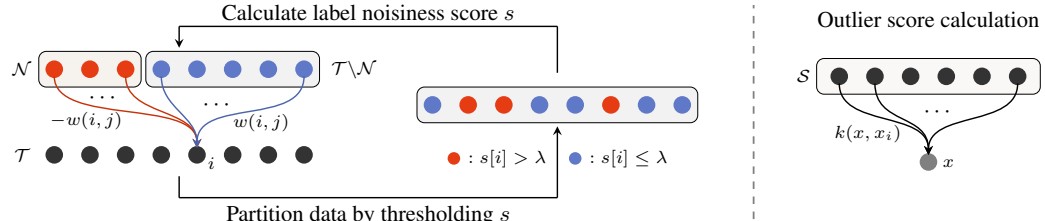

Figure 4: Illustration of our scoring algorithms for identifying label noise (left) and outliers (right).

correspond to any edges in the graph. Consistent with previous works [33], we aim to measure the *label noisiness score* for each data, where a higher score indicates a higher likelihood of label error. We denote the label noisiness scores for $\mathcal{T}$ as $s \in \mathbb{R}^n$, where $s[i]$ is the score for data $i$.

As depicted in Figure 3, data with label errors exhibit negative relations with other samples, implying that the data have similar features in the embedding space yet have dissimilarly assigned labels. This suggests that the edge weights $w(i, \cdot)$ quantify the extent to which the label assigned to node $i$ conflicts with the labels of other nodes. However, simply aggregating all edge weights of a node can yield suboptimal results, as negative relations can also contribute to the score for clean data, as shown in Figure 3. In Appendix D.2, we provide a more detailed experimental analysis of this issue.

To rectify this issue, we develop an algorithm that considers the global structure of the graph instead of simply summing the edge weights of individual nodes. Specifically, we identify subsets of data likely to have correct/incorrect labels and calculate the label noisiness score based on the subsets. We partition the nodes in $\mathcal{T}$ into two groups, where $\mathcal{N} \subset \mathcal{T}$ denotes the *estimated noisy subset* and $\mathcal{T} \setminus \mathcal{N}$ denotes the clean subset. To optimize $\mathcal{N}$, we aim to maximize the sum of the edges between the two groups, indicating that the label information of the two groups is the most conflicting. To ensure that $\mathcal{N}$ contains data with incorrect labels, which constitute a relatively small proportion of $\mathcal{T}$, we impose regularization to the cardinality of $\mathcal{N}$ with $\lambda \geq 0$ and formulate the following max-cut problem:

$$\mathcal{N}^* = \underset{\mathcal{N} \subset \mathcal{T}}{\operatorname{argmax}} \ \operatorname{cut}(\mathcal{N}, \mathcal{T} \setminus \mathcal{N}) \left( := \sum_{i \in \mathcal{N}} \sum_{j \in \mathcal{T} \setminus \mathcal{N}} w(i, j) \right) - \lambda |\mathcal{N}|. \tag{2}$$

The max-cut problem is NP-complete [11]. To solve this problem, we adopt the Kerninghan-Lin algorithm, which finds a local optimum by iteratively updating the solution [20]. However, the original algorithm that swaps data one by one at each optimization iteration is not suitable for large-scale settings. To this end, we propose an efficient *set-level* algorithm in Algorithm 1 that alternatively updates the noisy set $\mathcal{N}$ and label noisiness score vector $s$.

Specifically, given the current estimation of $\mathcal{N}$, the cut value excluding edges of node $i \in \mathcal{T}$ is $\operatorname{cut}(\mathcal{N} \setminus \{i\}, \mathcal{T} \setminus \mathcal{N} \setminus \{i\})$. Algorithm 1 measures the label noisiness score of node $i$ by comparing the objective cut values when including $i$ in $\mathcal{N}$ and when including $i$ in $\mathcal{T} \setminus \mathcal{N}$:

$$s[i] = \operatorname{cut}(\mathcal{N} \cup \{i\}, \mathcal{T} \setminus \mathcal{N} \setminus \{i\}) - \operatorname{cut}(\mathcal{N} \setminus \{i\}, \mathcal{T} \setminus \mathcal{N} \cup \{i\}) = \sum_{j \in \mathcal{T} \setminus \mathcal{N}} w(i, j) - \sum_{j \in \mathcal{N}} w(i, j).$$

Here we use the assumption $w(i, i) = 0$. In practice, the cardinality of $\mathcal{N}$ is small, so we can efficiently update the score vector $s$ by caching the initial score vector $\bar{s}$ as in Algorithm 1. After calculating the score vector $s$, we update the noisy set $\mathcal{N}$ by selecting nodes with score values above the value $\lambda$. Figure 4 illustrates the optimization process. Here larger values of $\lambda$ result in smaller $\mathcal{N}$ consisting of data samples that are more likely to have label noise. We provide the sensitivity analysis of $\lambda$ in Appendix C.1, with Table 7.

Algorithm 1 satisfies the convergence property in Proposition 1. In Appendix A, we provide proof and present an empirical convergence analysis on large-scale datasets. We also conduct a runtime

---

**Algorithm 1** Label noise identification

**Input:** Relation function $r \ (= -w)$
**Notation:** The number of data $n$
**for** $i = 1$ **to** $n$ **do**
  $\bar{s}[i] = \sum_{j=1}^{n} w(i, j)$   # caching initial score
**end for**
$s = \bar{s}$
**repeat**
  $\mathcal{N} = \{i \mid s[i] > \lambda, i \in [1, \dots, n]\}$
  **for** $i = 1$ **to** $n$ **do**
    $s[i] \leftarrow \bar{s}[i] - 2 \sum_{j \in \mathcal{N}} w(i, j)$
  **end for**
**until** convergence
**Output:** $s, \mathcal{N}$

---

analysis of our algorithm in Appendix A.3, demonstrating that the computation overhead of Algorithm 1 is negligible in large-scale settings.

**Proposition 1.** *Algorithm 1 with a single node update at each iteration converges to local optimum.*

**Complexity analysis** The time complexity of Algorithm 1 is $O(n^2)$, proportional to the number of edges in a graph. It is noteworthy that our method maintains the best performance when used with graphs consisting of a small number of nodes, as shown in Figure 5. This implies that we can partition large datasets and run the algorithm repeatedly for each partition to enhance efficiency while maintaining performance. In this case, the complexity becomes $O(n/k \cdot k^2) = O(nk)$, with $k$ representing the size of each partition and $n/k$ being the number of partitions. Also, computations on these partitions are embarrassingly parallelizable, meaning that the complexity becomes $O(k^2)$ for $k \ll n$ in distributed computing environments.

### 3.3 Outlier/OOD detection

In the previous section, we presented a method for detecting label errors based on data relations with similar feature embeddings but different label information. By employing the identical feature embedding structure, we identify outlier data by measuring the extent to which similar data are absent in the feature embedding space. To quantify the extent of a data point being an outlier, we aggregate the similarity kernel values of a data point in Equation (3), thereby processing the entire relational information of the data point. Our approach leverages global information about the data distribution, resulting in a more robust performance across a range of experimental settings compared to existing methods that rely on local information such as $k$-nearest distance [50]. Specifically, for a subset $\mathcal{S} \subseteq \mathcal{T}$ and data $x$, we measure the ***outlier score*** as

$$\text{outlier}(x) = \frac{1}{\sum_{i \in \mathcal{S}} k(x, x_i)}.$$

Higher values in the outlier score indicate that the data are more distributionally outliers. We propose to use a *uniform random* sampling for $\mathcal{S}$, adjusting the computational cost and memory requirements for the outlier score calculation to suit the inference environment. In Section 4.2, we verify our method maintains the best OOD detection performance even when using only 0.4% of the data in ImageNet.

### 3.4 Proposed similarity kernel

For $x_i \in \mathcal{X}$, we extract the feature representation $\mathbf{f}_i$ and the prediction probability vector $\mathbf{p}_i$ from the trained model. We propose a class of bounded kernel $k : \mathcal{X} \times \mathcal{X} \to [0, M]$ with the following form:

$$k(x_i, x_j) = |s(\mathbf{f}_i, \mathbf{f}_j) \cdot c(\mathbf{p}_i, \mathbf{p}_j)|^t. \tag{3}$$

A positive scalar value $t$ controls the sharpness of the kernel value distribution. A larger value of $t$ makes a small kernel value smaller, which is effective in handling small noisy kernel values. A scalar value $s(\mathbf{f}_i, \mathbf{f}_j) \in \mathbb{R}^+$ denotes a similarity measurement between features. In our main experiments, we adopt the truncated cosine-similarity that has been widely used in representation learning [44, 42]. We use the hinge function at zero, resulting in the following positive feature-similarity function:

$$s(\mathbf{f}_i, \mathbf{f}_j) = \max(0, \cos(\mathbf{f}_i, \mathbf{f}_j)).$$

It is worth noting the utility of our framework is not limited to a specific kernel design. In Section 4.3, we verify our approach maintains the best performance with $s(\mathbf{f}_i, \mathbf{f}_j)$ defined as the RBF kernel [60].

While the feature similarity captures the meaningful semantic relationship between data points, we observe that considering the prediction scores $\mathbf{p}_i$ can further improve the identification of problematic data. To incorporate prediction scores into our approach, we introduce a scalar term $c(\mathbf{p}_i, \mathbf{p}_j)$ that measures the compatibility between the predictions on data points. Any positive and bounded compatibility function is suitable for the kernel class defined in Equation (3). In our main experiments, we use the predicted probability of belonging to the same class as the compatibility term $c(\mathbf{p}_i, \mathbf{p}_j)$. Specifically, given the predicted label random variables $\hat{y}_i$ and $\hat{y}_j$, the proposed compatibility term is

$$c(\mathbf{p}_i, \mathbf{p}_j) = P(\hat{y}_i = \hat{y}_j) = \mathbf{p}_i^\mathsf{T} \mathbf{p}_j. \tag{4}$$

From a different perspective, we interpret this term as a measure of confidence for feature similarity. In Section 4.3, we verify the effectiveness of the compatibility term through an ablation study.

**Interpretation** To better understand our relation function with kernel defined in Equation (3), we draw a connection to the influence function [39], which estimates the influence between data points by computing the inner product of the network gradient on the loss function $\ell$ of each data point as $\nabla_w \ell(x_i)^\intercal \nabla_w \ell(x_j)$, where $w$ denotes the network weights. Following the convention, we consider an influence function on the feed-forward layer, where $\ell(x_i) = h(\mathbf{f}'_i) = h(w^\intercal \mathbf{f}_i)$. By the chain rule, we can decompose the network gradient as $\nabla_w \ell(x_i) = \nabla_{\mathbf{f}'} h(\mathbf{f}'_i) \mathbf{f}_i^\intercal$, and represent the influence as $\nabla_{\mathbf{f}'} h(\mathbf{f}'_i)^\intercal \nabla_{\mathbf{f}'} h(\mathbf{f}'_j) \cdot \mathbf{f}_i^\intercal \mathbf{f}_j$. Our relation function differs from the influence function in that it does not rely on feature gradients $\nabla_{\mathbf{f}'} h(\mathbf{f}'_i)$ to evaluate the relationship between data points. Instead, our relation function compares model predictions and assigned labels independently using the terms $c(\mathbf{p}_i, \mathbf{p}_j)$ and $1(y_i, y_j)$. Our formulation does not require computationally expensive back-propagation and more robustly identifies conflicting data information than influence functions which are known to be sensitive to outliers [2]. We provide a more detailed theoretical analysis in Appendix A.4.

## 3.5 Data relation map

In this section, we present a visualization method based on our data relation function to contextualize data and comprehend its relational structure. One of the effective approaches for visualizing a dataset is dataset cartography [51], which projects the dataset onto a 2D plot. This approach draws a scatter plot of the mean and standard deviation of the model's prediction probabilities for each data sample during training. Inspired by the dataset cartography, we propose a *data relation map*, which visualizes the relationship between data along the training process. To this end, we uniformly store checkpoints during training. We denote a set of these checkpoints as $\mathcal{K}$, where $r_k$ refers to the relation function for checkpoint $k \in \mathcal{K}$. For each data sample $i \in \mathcal{T}$, we draw a scatter plot of the mean and standard deviation of relation values $\{r_k(i,j) \mid k \in \mathcal{K}\}$ for $j \in \mathcal{T} \setminus \{i\}$.

In Figure 1, we provide relation maps of three samples from ImageNet, using 10 checkpoints of MAE-Large [13]. The three samples each represent clean data, data with a label error, and outlier data. From the figure, samples show different relation map patterns. Specifically, the relation map of a clean data sample exhibits a majority of positive relations with relatively small variability. We note that there are gray-colored relations in high variability regions (0.2<std), indicating that the model resolves conflicting relations at convergence. On the other hand, the relation map of the sample with a label error demonstrates a majority of negative relations. Notably, high variance relations result in largely negative relations at convergence, suggesting that conflicts intensify. Lastly, the relation map of the outlier data sample reveals that relations are close to 0 during training. These relation maps can serve as a model-based fingerprint of the data, which our algorithm effectively exploits to identify problematic data. We provide additional data relational maps for various models in Appendix E.1.

# 4 Experimental results

In this section, we experimentally verify the effectiveness of our approach in detecting label errors and outliers. We provide implementation details, including hyperparameter settings in Appendix C.1. We provide qualitative results including detected label errors and outlier samples in Appendix E.2.

## 4.1 Label error detection

### 4.1.1 Setting

**Datasets** We conduct label error detection experiments on large-scale datasets: ImageNet [43], ESC-50 [37], and SST2 [55]. ImageNet consists of about 1.2M image data from 1,000 classes. ESC-50 consists of 2,000 5-second environmental audio recordings organized into 50 classes. SST2 is a binary text sentiment classification dataset, consisting of 67k movie review sentences. We also conduct experiments on MNLI [55] and provide results in Appendix, Table 13.

Following Pruthi et al. [39], we construct a noisy training set by flipping labels of certain percentages of correctly classified training data with the top-2 prediction of the trained model. We use different neural network architectures for constructing a noisy training set and detecting label errors to avoid possible correlation. We leave a more detailed procedure for constructing the noisy training set in Appendix C.2. When training the MAE-Large model [13] on ImageNet with 8% label noise, the validation top1-accuracy decreases by 1.7% compared to the performance of the model trained on the orig-

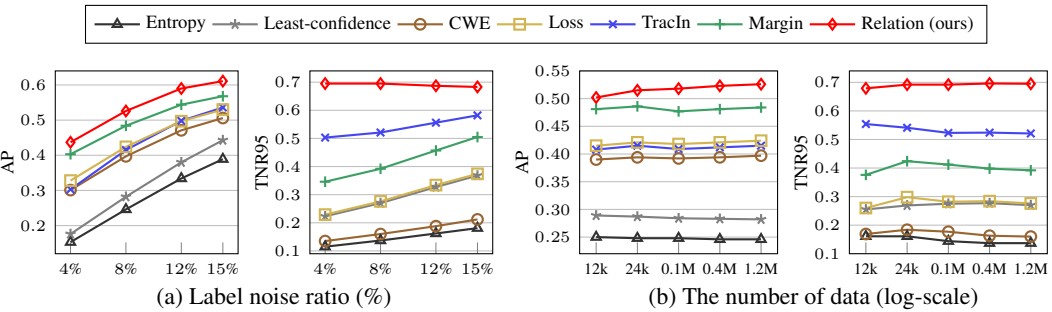

(a) Label noise ratio (%)                    (b) The number of data (log-scale)

Figure 5: Label error detection performance on ImageNet with MAE-Large according to (a) label noise ratios and (b) the number of data. We obtain the results in (b) with 8% label noise. We report performance values of all methods in Appendix D.3, with Tables 10 and 11.

inal training set, as reported in Appendix Table 8. The decrease becomes more significant with 15% label noise, with a drop of 4.4%, highlighting the importance of label noise detection and cleaning.

**Baselines**   We compare our method (*Relation*) to six baselines that are suitable for large-scale datasets. We consider fine-tuned loss from pre-trained models (*Loss*) [5], prediction probability margin score (*Margin*) [33], and the influence-based approach called *TracIn* [39]. We also evaluate model-agnostic scoring methods: *Entropy*, *Least-confidence*, and Confidence-weighted Entropy (*CWE*) [24]. For a fair comparison, we evaluate methods using a single converged neural network in our main experiments, while also providing results with a temporal ensemble suggested by [39] in Appendix D.1.

**Metric**   We evaluate the detection performance based on label noisiness scores by each method. We note that detecting label errors is an imbalanced detection problem, which makes the AUROC metric prone to being optimistic and misleading [7]. In this respect, we mainly report the AP (average precision) and TNR95 (TNR at 0.95 TPR), and provide AUROC results in Appendix D.3.

#### 4.1.2   Results and analysis

**ImageNet**   We measure the label error detection performance on ImageNet with the synthetic label noise by training an MAE-Large model [13]. Note that the model does not have access to information about the changed clean labels during the entire training process. Figure 5 (a) shows the detection performance over a wide range of label noise ratios from 4% to 15%. As shown in the figure, our approach achieves the best AP and TNR95 performance compared to the baselines. Especially, our method maintains a high TNR95 over a wide range of noise ratios, indicating that the number of data that need to be reviewed by human annotators is significantly smaller when cleaning the dataset. In Figure 9, we present detected label error samples by our algorithm.

It is worth noting that our method relies on the number of data for constructing a relation graph. To measure the sensitivity of our algorithm to the number of data, we evaluate the detection performance using a reduced number of data with uniform random sampling. Figure 5 (b) shows the detection performance on 8% label noise with MAE-Large. From the figure, we find that our algorithm maintains the best detection performance even with 1% of the data (12k). This demonstrates that our algorithm is effective even when only a small portion of the training data is available, such as continual learning or federated learning [35, 32]. In Table 1 (a), we provide detection performance for different scales of MAE models on 8% label noise. The table shows our approach achieves the best AP with MAE-Base, verifying the robustness of our approach to the network scales. From the table, we note that larger models are more robust to label noise and show better detection performance.

**Speech and language domains**   We apply our method to speech and language domain datasets: ESC-50 [37] and SST2 [55]. We design the label noise detection settings identical to the previous ImageNet section. Specifically, we train the AST model [10] for ESC-50 and the RoBERTa-Base model [30] for SST2 under the 10% label noise setting. Table 1 (b) shows our approach achieves the best AP and TNR95 on the speech and language datasets, demonstrating the generality of our approach across various data types.

Table 1: Label error detection performance on ImageNet with 8% label noise. *Baseline* refers to the *best* performance among the six baselines considered in Figure 5. In Table (c), the evaluation metric is AP. We report the performance of all baselines in Appendix D.3, with Tables 12 to 14.

(a) Model architecture scales

| Scale | Metric | Baseline | Relation |
|---|---|---|---|
| Base | AP | 0.477 | **0.514** |
| | TNR95 | 0.488 | **0.672** |
| Large | AP | 0.484 | **0.526** |
| | TNR95 | 0.521 | **0.695** |

(b) Speech/language domains

| Dataset | Metric | Baseline | Relation |
|---|---|---|---|
| ESC50 | AP | 0.739 | **0.779** |
| | TNR95 | 0.793 | **0.847** |
| SST2 | AP | 0.861 | **0.881** |
| | TNR95 | 0.850 | **0.870** |

(c) Realistic label noise scenario

| Model | Baseline | Relation |
|---|---|---|
| MAE | 0.708 | **0.733** |
| BEIT | 0.719 | **0.737** |
| ConvNeXt | 0.713 | **0.735** |
| ConvNeXt-22k | 0.724 | **0.744** |

**Realistic label noise**    The ImageNet validation set is known to contain numerous label errors [34]. To tackle this issue, Beyer et al. [4] cleaned the labels with human experts and corrected around 29% of the labels via multi-labeling. With this re-labeled validation set, we conduct experiments under the realistic label noise, with the task of detecting the data samples with changed labels. We measure the detection performance with MAE-Large [13], BEIT-Large [1], and ConvNeXt-Large [31] models. To examine the impact of pre-training on external data, we also include ConvNeXt pre-trained on ImageNet-22k, denoted as ConvNeXt-22k. We construct the relation graph using only the validation set, considering scenarios where the training data are not available. Table 1 (c) verifies that our approach outperforms the best baseline across various models. The results on ConvNeXt-22k indicate that pre-training on external data improves the detection performance.

**Memorization issue**    We investigate the impact of large neural networks' ability to memorize label errors on detection performance [61]. In the left figure of Figure 6, we find that the AP score decreases as the training progresses after 30 epochs with MAE-Large which converges at 50 epochs. The right figure of Figure 6 plots the precision-recall curves, where we observe that precision increases at low recall area but decreases at mid-level recall ($\sim 0.5$) as the training progresses. This suggests that training has both positive and negative effects on detecting label noise, and we speculate that memorization

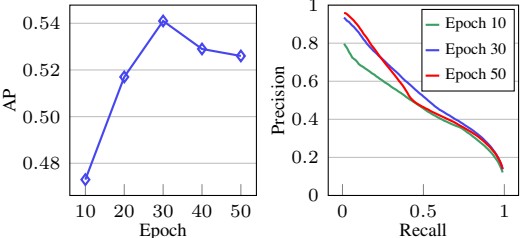

Figure 6: Label error detection performance of relation graph throughout the MAE-Large training process on ImageNet with 8% label noise.

is one cause. Leveraging these observations, we improve detection AP by 3.6%p by using a temporal ensemble of models [25]. We provide a more detailed description and results in Appendix D.1.

## 4.2    Outlier/OOD detection

**Baselines**    We consider the following representative outlier scoring approaches: Maximum Softmax Probability (*MSP*) [14], *Max Logit* [15], *Mahalanobis* [26], *Energy* score [29], *ReAct* [49], *KL-Matching* [15], and *KNN* [50]. We tune the KNN method's hyperparameter $k$ based on the paper's guidance as $k = 1000 \times \alpha$, where $\alpha$ represents the ratio of training data used for OOD detection. We also evaluate outlier detection approaches, *Iterative sampling* [48] and *Local outlier factor* [56].

**OOD detection**    Following Sun et al. [50], we evaluate OOD detection performance on the ImageNet validation set consisting of 50k in-distribution data samples, along with four distinct OOD datasets: *Places* [62], *SUN* [58], *iNaturalist* [53], and *Textures* [6]. Each of these OOD datasets consists of 10k data samples except for Textures which has 5,640 data samples. We also combine these four datasets, denoted as *ALL*, and measure the overall OOD detection performance on this dataset.

Figure 7 shows OOD detection performance of MAE-Large on ALL outlier dataset. Note that our approach and KNN both rely on the number of training data samples ($|\mathcal{S}|$) for outlier score calculation. We examine the effect of training set size by measuring the performance with a reduced number of data using uniform random sampling. Figure 7 verifies that our approach outperforms other baselines while maintaining performance even with 0.4% of the training dataset (5k). Note that KNN requires hyperparameter tuning according to the training set size, whereas our approach uses the identical

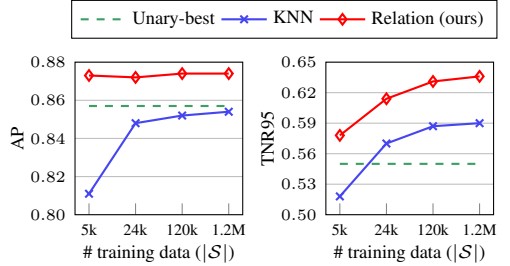
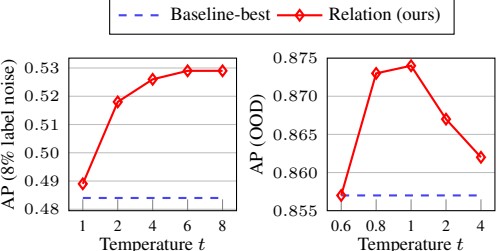

Figure 7: OOD detection performance on ImageNet (ALL) with MAE-Large. *Unary-best* refers to the best performance among the methods that do not rely on the training data for outlier score calculation. We provide performance values of all baselines in Appendix D.4, with Table 15.

Figure 8: The detection AP of the MAE-Large model across a range of kernel temperatures $t$. The dashed blue line means the performance of the best baseline.

Table 2: Outlier detection performance with Vit-Base on noisy ImageNet-100. Some OOD scoring methods (Mahalanobis, ReAct, KL-Matching) are excluded from the comparison because they require a clean training dataset which is not available in the outlier detection setup.

<table>
<tr><td colspan="4">(a) ImageNet-100 with SUN</td><td colspan="4">(b) ImageNet-100 with iNaturalist</td></tr>
<tr><th>Method</th><th>AUROC</th><th>AP</th><th>TNR95</th><th>Method</th><th>AUROC</th><th>AP</th><th>TNR95</th></tr>
<tr><td>MSP</td><td>0.708</td><td>0.335</td><td>0.032</td><td>MSP</td><td>0.706</td><td>0.309</td><td>0.040</td></tr>
<tr><td>Max Logit</td><td>0.499</td><td>0.216</td><td>0.011</td><td>Max Logit</td><td>0.469</td><td>0.171</td><td>0.012</td></tr>
<tr><td>Energy</td><td>0.417</td><td>0.106</td><td>0.010</td><td>Energy</td><td>0.375</td><td>0.075</td><td>0.012</td></tr>
<tr><td>KNN</td><td>0.990</td><td>0.899</td><td>0.960</td><td>KNN</td><td>0.993</td><td>0.923</td><td>0.972</td></tr>
<tr><td>Iterative sampling</td><td>0.973</td><td>0.687</td><td>0.903</td><td>Iterative sampling</td><td>0.979</td><td>0.734</td><td>0.922</td></tr>
<tr><td>Local outlier factor</td><td>0.986</td><td>0.850</td><td>0.941</td><td>Local outlier factor</td><td>0.990</td><td>0.890</td><td>0.958</td></tr>
<tr><td>Relation (Ours)</td><td>**0.993**</td><td>**0.906**</td><td>**0.971**</td><td>Relation (Ours)</td><td>**0.995**</td><td>**0.940**</td><td>**0.982**</td></tr>
</table>

hyperparameter ($t = 1$) regardless of the size. In Appendix D.4, Tables 15 and 16, we provide OOD detection results on four individual OOD datasets as well as the performance with ResNet-50 [12], where our approach achieves the best OOD detection performance over the nine baselines considered.

**Outlier detection** We perform outlier detection experiments following the methodology by Wang et al. [56], where the training set contains outlier data with random labels. We construct the noisy ImageNet-100 training sets by using iNaturalist [53] and SUN [58] datasets. We train a ViT-Base model [8] from scratch on these noisy training datasets, and measure outlier detection performance using the trained model. For a more detailed description, please refer to Appendix C.3.

Table 2 shows the outlier detection results on two outlier datasets. As indicated, our method achieves the best performance in both outlier settings, demonstrating its effectiveness in outlier detection. It is worth noting that the considered OOD scoring methods (MSP, Max Logit, Energy) do not achieve good outlier detection performance. We speculate that this is due to the overfitting of the neural network's predictions on outliers.

**Detecting outliers in validation set** We further utilize our method for identifying outliers in the validation set by retrieving data samples with the highest outlier score (Section 3.3). In Figure 10, we present samples detected by our algorithm from ImageNet and SST2. In the figure, we observe that these samples are not suitable for measuring the predictive performance on labels, which should be excluded from the evaluation dataset.

## 4.3 Ablation study

**Temperature** $t$ In Equation (3), we introduced a temperature $t$, where a large value of $t$ increases the influence of large relation values in our algorithm. We conduct sensitivity analysis on $t$ with MAE-Large on ImageNet under 8% label noise. Figure 8 shows the effect of the temperature value

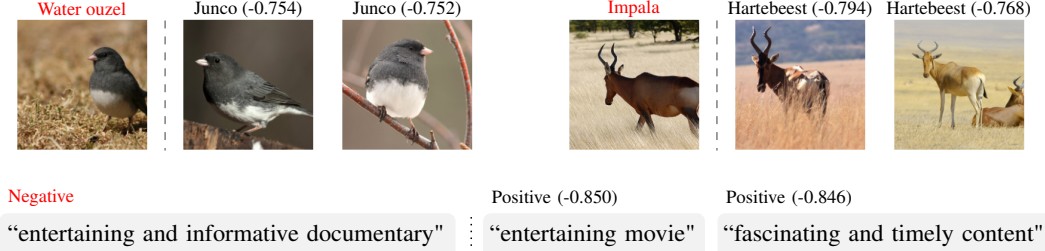

Figure 9: Detected data samples with label errors (marked in red) from ImageNet (top) and SST2 (bottom). We present samples with conflicting relations next to the detected samples and denote the corresponding relation value in parenthesis. We present more samples in Appendix E.2, Figure 14.

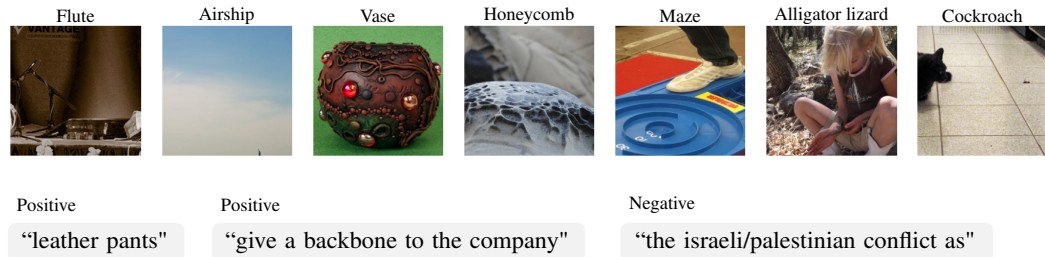

Figure 10: Data samples with the highest outlier scores by our method on ImageNet (top) and SST2 (bottom) validation sets. We denote the assigned labels above each data sample. We present more outlier samples in Appendix E.2, Figure 15 and Table 18.

on our detection algorithm's performance. From the figure, we observe that the label error detection performance increases as the $t$ value increases, saturating at $t = 6$. In the case of OOD detection, we achieve the best performance at around $t = 1$. Our algorithm outperforms the best baseline over a wide range of hyperparameters, demonstrating the robustness of our algorithm to the hyperparameter.

**Similarity kernel design**  We present an empirical analysis of the kernel design choices. Specifically, we replace the cosine similarity term in Equation (3) as the RBF kernel and evaluate the detection performance. We further conduct an ablation study on compatibility terms (Equation (4)). Table 3 summarizes the label error detection performance with different kernel functions on ImageNet with 8% noise ratio. The table shows that our approach largely outperforms the best baseline even with the RBF

Table 3: Comparison of similarity kernel designs. *Baseline* represents the best baseline performance. The term $c$ denotes our compatibility term in Equation (4). Note, $Cos \cdot c$ is the kernel function considered in our main experiments, and *RBF / Cos* refers to our method without the compatibility term $c$.

| Metric | Baseline | RBF | Cos | RBF $\cdot$ $c$ | Cos $\cdot$ $c$ |
|--------|----------|-------|-------|-------------|-------------|
| AP     | 0.484    | 0.470 | 0.471 | 0.525       | 0.526       |
| TNR95  | 0.521    | 0.668 | 0.671 | 0.703       | 0.695       |

kernel. Also, we find that our approach without the compatibility term shows comparable AP performance while significantly outperforming baselines in TNR95. These results demonstrate the generality and utility of our relational structure-based framework, which is not limited to a specific kernel design.

## 5   Conclusion

In this paper, we propose a novel data relation function and graph algorithms for detecting label errors and outlier data using the relational structure of data in the feature embedding space. Our approach achieves state-of-the-art performance in both label error and outlier/OOD detection tasks, as demonstrated through extensive experiments on large-scale benchmarks. Furthermore, we introduce a data contextualization tool based on our data relation that can aid in data diagnosis. Our algorithms and tools can facilitate the analysis of large-scale datasets, which is crucial for the development of robust machine-learning systems.

## Acknowledgement

We are grateful to Jinuk Kim for helpful discussions. This work was supported by SNU-NAVER Hyperscale AI Center, Institute of Information & Communications Technology Planning & Evaluation (IITP) grant funded by the Korea government (MSIT) [No. 2020-0-00882, (SW STAR LAB) Development of deployable learning intelligence via self-sustainable and trustworthy machine learning and NO.2021-0-01343, Artificial Intelligence Graduate School Program (Seoul National University)], and Basic Science Research Program through the National Research Foundation of Korea (NRF) funded by the Ministry of Education (RS-2023-00274280). Hyun Oh Song is the corresponding author.

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

# A Algorithm analysis

## A.1 Proof

In this section, we provide proof for Proposition 1. Recall that Algorithm 1 updates an estimated noisy set $\mathcal{N}$ at a set-level as $\mathcal{N} = \{i \mid s[i] > \lambda\}$. We can conduct Algorithm 1 with a single node update at each iteration using the criterion $s[i] > \lambda$, referred to as a sample-level version of Algorithm 1. Specifically, let us define $v \in \mathbb{R}^n$ such that $v[i] = -1$ for $i \in \mathcal{N}$ and $v[i] = 1$ for else. Then the algorithm moves a sample $k$ to another partition at each iteration, where $k = \operatorname{argmax}_{i \in \mathcal{T}} v[i](s[i] - \lambda)$. The algorithm stops when $v[k](s[k] - \lambda) \leq 0$.

**Proposition 1.** *Algorithm 1 with a single node update at each iteration converges to local optimum.*

*Proof.* The change in the objective value of Equation (2) by moving data $i$ from $\mathcal{T} \backslash \mathcal{N}$ to $\mathcal{N}$ is

$$\sum_{j \in \mathcal{T} \backslash \mathcal{N}} w(i,j) - \sum_{j \in \mathcal{N}} w(i,j) - \lambda, \tag{5}$$

where the change by moving data $i$ from $\mathcal{N}$ to $\mathcal{T} \backslash \mathcal{N}$ is

$$\sum_{j \in \mathcal{N}} w(i,j) - \sum_{j \in \mathcal{T} \backslash \mathcal{N}} w(i,j) + \lambda.$$

Note the score $s$ in Algorithm 1 is

$$s[i] = \sum_{j \in \mathcal{T}} w(i,j) - 2 \sum_{j \in \mathcal{N}} w(i,j) = \sum_{j \in \mathcal{T} \backslash \mathcal{N}} w(i,j) - \sum_{j \in \mathcal{N}} w(i,j).$$

Thus the change in the objective value by moving data $i$ to another partition is $s[i] - \lambda$ for $i \in \mathcal{T} \backslash \mathcal{N}$ and $-s[i] + \lambda$ for $i \in \mathcal{N}$, which can be represented as $v[i](s[i] - \lambda)$. Therefore, moving a sample with a positive value of $v[i](s[i] - \lambda)$ to another partition guarantees an increase in the objective function value. Because a cut value in a graph is bounded, the algorithm converges to the local optimum by the monotone convergence theorem. $\square$

## A.2 Empirical convergence analysis

We conduct an empirical study on the convergence of Algorithm 1. Specifically, we randomly sample 100,000 data from ImageNet and construct a relation graph. We compare the set-level Algorithm 1 and the original Kerninghan-Lin algorithm in Figure 11. The figure indicates that both algorithms converge to the local optimum, while our set-level algorithm converges faster. Additionally, we observe that the set-level algorithm achieves a lower objective value, verifying its effectiveness in large-scale settings.

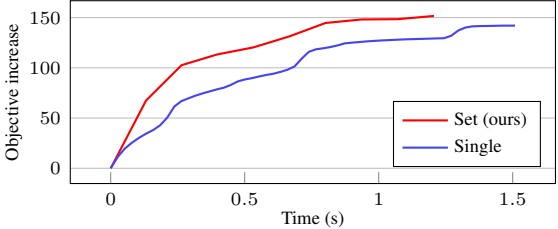

Figure 11: Empirical convergence analysis of max-cut algorithms. *Set* denotes Algorithm 1 and *Single* denotes the Kerninghan-Lin algorithm which updates a single node at each optimization step.

## A.3 Computation time comparison

In this section, we measure the time spent on detection algorithms. We use 1 RTX3090-Ti GPU and conduct experiments on the full ImageNet training set. Table 4 compares computation time for Algorithm 1 and feature calculation. Note that all existing methods based on neural networks, including ours, require the calculation of features. The table shows that Algorithm 1 (excluding

Table 4: Time spent (s) for label error detection on ImageNet 1.2M dataset. *Feature* indicates the total computing time for calculating feature embeddings of all data points, and *Gradient* means the total computing time for calculating network gradient on each data point. *Algorithm 1* indicates the time spent by our algorithm, excluding feature calculation.

| Model | Feature | Algorithm 1 | Gradient |
|-------|---------|-------------|----------|
| MAE-Base | 2300 | 400 | 6000 |
| MAE-Large | 6900 | 420 | 21000 |

Table 5: Time spent per sample (ms) for OOD detection on ImageNet with various models. *Unary* refers to unary scoring methods utilizing logit or probability score for each data point.

| Model | Unary | Relation |
|-------|-------|----------|
| MAE-Large | 12.2 | 12.3 |
| ResNet-50 | 8.1 | 8.2 |

feature calculation) requires significantly less computation time than forward computation. We also observe that our algorithm efficiently scales up to larger neural networks which have a larger number of feature embedding dimensions. It is also worth noting that computing gradient takes a much longer time and also requires a large memory budget, demonstrating the efficiency of our algorithm in large-scale label error detection.

In Table 5, we measure the time spent for OOD detection on the full ImageNet training set. The computation of our similarity kernel is embarrassingly parallelizable on GPUs. As shown in the table, the overhead time for computing our outlier scores is negligible compared to the time spent for the neural networks' forward computation on a single data point. We can further reduce the time and memory requirements by measuring the outlier score on a subset of the training set (Figure 7).

### A.4   Interpretation of relation function

We establish an understanding of our relation function in Equation (1) by drawing a connection to the influence function [39]. For simplicity, we consider the influence function with a single checkpoint, where the influence between data $x_i$ and $x_j$ is given by $\nabla_w \ell(x_i)^\intercal \nabla_w \ell(x_j)$. Here, $\ell$ denotes the loss function, and $w$ denotes the weight of the checkpoint. We consider the influence function at the feed-forward layer, where $\ell(x_i) = h(\mathbf{f}_i') = h(w^\intercal \mathbf{f}_i)$, following the convention [39]. By the chain rule, we can decompose the weight gradient as $\nabla_w \ell(x_i) = \nabla_{\mathbf{f}'} h(\mathbf{f}_i') \mathbf{f}_i^\intercal$, and represent the influence as $\nabla_{\mathbf{f}'} h(\mathbf{f}_i')^\intercal \nabla_{\mathbf{f}'} h(\mathbf{f}_j') \cdot \mathbf{f}_i^\intercal \mathbf{f}_j$. In contrast, our relation function has a form of $1(y_i = y_j) \cdot |s(\mathbf{f}_i, \mathbf{f}_j) \cdot c(\mathbf{p}_i, \mathbf{p}_j)|^t$.

The main distinction between our relation function and the influence function is the existence of the feature gradient term $\nabla_{\mathbf{f}'} h(\mathbf{f}_i')$. As observed in Barshan et al. [2], outliers have a large feature-gradient norm, leading to difficulties in detecting label errors. Specifically, let us consider the weight $w$ at the classifier layer, where the function $h$ is the softmax cross-entropy loss function. As Pruthi et al. [39], we can express the feature gradient inner-product as

$$\nabla_{\mathbf{f}'} h(\mathbf{f}_i')^\intercal \nabla_{\mathbf{f}'} h(\mathbf{f}_j') = (\mathbf{y}_i - \mathbf{p}_i)^\intercal (\mathbf{y}_j - \mathbf{p}_j),$$

where $\mathbf{y}_i$ denote the one-hot label. The equation above shows that the correctly classified data with $\mathbf{y}_i \approx \mathbf{p}_i$ yields near zero inner-product values, whereas outliers with high entropy predictions exhibit large inner-product values. Consequently, existing influence-based label error detection methods, which detect label errors by identifying data with high influence values, have degraded performance in the presence of outliers [2].

Our relation function differs from influence functions in that it separates label and prediction information using a label comparison term $1(y_i = y_j)$ and a compatibility term $c(\mathbf{p}_i, \mathbf{p}_j)$, respectively. Outlier data typically have a high entropy of model predictions, resulting in lower compatibility values with other data [14]. On the other hand, normal data with label errors exhibit high compatibility values with other normal data. Our detection algorithms exploit these differences and achieve improved detection performance compared to the influence functions.

## B   Additional discussions

**Why does relation graph work?**   We discuss the conceptual differences between our relation graph-based approach and previous baselines. Firstly, our method is data-centric, whereas the previous

Table 6: Hyperparameter settings. We use identical hyperparameters for all experiments of each task regardless of model types.

| Task | $t$ (Equation (3)) | $\lambda$ (Algorithm 1) |
|---|---|---|
| Label error | 4 | 0.05 |
| Outlier | 6 | - |
| OOD | 1 | - |

Table 7: Label error detection performance over a range of $\lambda$ values (ImageNet 8% noise, MAE-Large).

| Metric \ $\lambda$ | 0.0 | 0.01 | 0.05 | 0.1 |
|---|---|---|---|---|
| AP | 0.522 | 0.524 | 0.527 | 0.527 |
| TNR95 | 0.692 | 0.693 | 0.695 | 0.693 |

approaches rely on a unary score by models. Our approach identifies problematic data by comparing them to other data, which leads to more reliable identification of problematic data than unary scoring methods that are vulnerable to overfitting [38]. Secondly, our approach aggregates global relational information, whereas previous methods rely on local information such as $k$-nearest distance [50]. Considering all edge connections, our method obtains more representative information about the data distribution. Through temperature parameter $t$ and efficient graph algorithms, we effectively process the entire relations and achieve the improved identification of problematic data.

**Limitations and future works**   There are several promising future directions for our work. Firstly, the current experiments are limited to the classification task, and it would be valuable to apply our approach to a wider range of tasks, such as segmentation or generative models. These tasks may introduce new and interesting categories of problematic data arising from different label spaces and data structures. Secondly, integrating our method with human annotation and model training processes will also be valuable. This could involve using our approach to identify inconsistencies in label assignments or to conduct a fine-grained evaluation of models.

**Broader impact**   Regarding social impact, we anticipate that our method will largely reduce the cost of the data cleaning and annotation process, while also providing developers with valuable insights into their data and training process through our data relation map. We do not anticipate negative social effects, as our method does not deal with socially sensitive issues.

## C   Experiment settings

### C.1   Implementation details

**Models**   For label error detection, we train models on datasets with label noise. In the case of ImageNet, we fine-tune the pre-trained MAE models following the official training codes[2], which train MAE-Large for 50 epochs and MAE-Base for 100 epochs. The MAE-Large model has a feature dimension of 1024, where the MAE-Base model has a dimension of 768. It is worth noting that the masked auto-encoding pre-training process of MAE does not utilize label information. We use 8 RTX3090-Ti GPUs for the training and 1 RTX3090-Ti GPU for executing our algorithm. In the case of ESC-50, we fine-tune the AST model with a 768 feature dimension for 25 epochs following the official training codes[3]. In the case of language domain tasks, we fine-tune RoBERTa-Base with a 768 feature dimension following the official training codes[4], where we train models for 5 epochs. For other models, we use the trained models provided by the Timm library[5]. For all experiments, we use the inputs of the classification layers as feature embeddings. In the case of RoBERTa, this corresponds to the encoder output of the [CLS] token.

**Hyperparameter**   As shown in Figure 8, we observe that a large value of temperature $t$ benefits label error detection, while a moderate temperature value around 1 shows the best performance on OOD detection. We summarized the hyperparameter used in our experiments in Table 6. Note we use $\lambda = 0.05$ in Algorithm 1 after scaling the label noisiness score to have a maximum absolute value of 1. We observe that the conservative estimation of noisy set $\mathcal{N}$ by using small positive $\lambda$ values leads to better results than $\lambda = 0$, as shown in Table 7. The results in the table confirm that our method

---

[2]`https://github.com/facebookresearch/mae`

[3]`https://github.com/YuanGongND/ast`

[4]`https://github.com/facebookresearch/fairseq/tree/main/examples/roberta`

[5]`https://github.com/rwightman/pytorch-image-models`

Table 8: Validation top-1 accuracy of MAE-Large trained on ImageNet with noisy labels.

| Label Noise Ratio | 0. | 0.04 | 0.08 | 0.12 | 0.15 |
|---|---|---|---|---|---|
| Top-1 Accuracy | 85.89 | 84.96 | 84.15 | 82.88 | 81.50 |

Table 9: Label error detection AP by the temporal model ensemble with 4 MAE-Large checkpoints (ImageNet, 8% label noise). In parenthesis, we denote the performance gain compared to the detection by a single converged model.

| Entropy | Least-conf. | CWE | Loss | TracIn | Margin | Relation |
|---|---|---|---|---|---|---|
| 0.246 (0.007) | 0.282 (0.001) | 0.397 (0.031) | 0.465 (0.041) | 0.449 (0.034) | 0.544 (0.06) | **0.562** (0.036) |

performs effectively over a wide range of $\lambda$ values, outperforming the best scores of 0.484 AP and 0.521 TNR95 from the six baselines tested.

**Other tricks**  We find that small noisy kernel values accumulate errors as we consider large numbers of data. To resolve this issue, we clamp small similarity kernel values that fall below an absolute value of 0.03 as zero in Equation (3).

## C.2  Synthetic label noise

In Section 4.1, we conduct controlled experiments by generating synthetic label noise. Specifically, we flip labels of a certain percentage of training data with the top-2 prediction of trained models on correctly classified data. For the label flip, we use MAE-Huge for ImageNet and RoBERTa-Large for language datasets. Note that we did not use these models for detecting label errors to prevent possible correlations. In the case of speech domain, we use the identical AST architecture. We note that the original ImageNet training set may contain label issues which can lead to misleading experimental results [34]. Therefore, we remove about 4% of data that are likely to have label issues by following Northcutt et al. [33] with MAE-Huge, resulting in a total of 1,242,890 data samples. We conduct label error detection experiments on this pre-cleaned training set. In Table 8, we provide the top-1 validation accuracy of MAE-Large trained on training sets with noisy labels, demonstrating the importance of label noise identification and cleaning.

## C.3  Outlier detection setting

As outlier detection is not well-benchmarked in modern computer vision [59], we design an experimental setup with the ImageNet dataset and two outlier datasets: SUN [58] and iNaturalist [53], each with 10k outlier data. To ensure an appropriate outlier data ratio ($\sim$8%), we use ImageNet-100 [21], a subset of ImageNet consisting of 120k data from 100 classes. We adopt the outlier detection setting of Wang et al. [56], where the training set has the outlier data with random labels. To consider multiple types of outlier data, we construct two noisy training sets: ImageNet-100 with SUN and ImageNet-100 with iNaturalist. We train a ViT-Base model [8] from scratch for 300 epochs on these noisy training datasets and measure outlier detection performance using the trained model.

## D  Additional experimental results

### D.1  Temporal model ensemble

Following Pruthi et al. [39], we measure the label error detection performance by using the temporal model ensemble. Specifically, we average the label noisiness scores from 4 checkpoints that are uniformly sampled throughout training. Table 9 shows that this technique improves the performance of all methods, with our approach still exhibiting the best performance. These results confirm the effectiveness of temporal ensembles when more computation and storage are available.

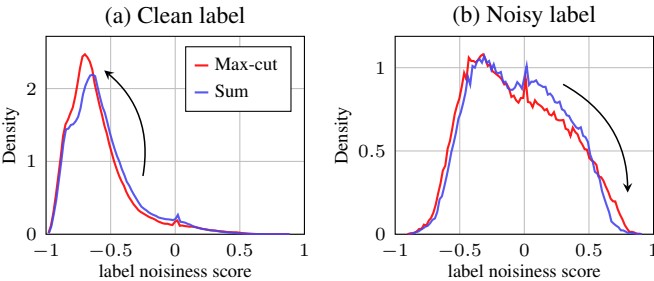

Figure 12: Normalized histogram of label noisiness scores by simple edge aggregation (*Sum*) and max-cut. Arrows indicate differences between the density functions.

### D.2 Analysis on edge aggregation

In this section, we analyze the difference between label noisiness scores calculated using max-cut and simple edge sum. Specifically, in Figure 12, we draw histograms of label noisiness scores for two groups of data: one with clean labels and the other with label errors. The figure shows that the score calculated through max-cut shifts the density in a positive direction for data with clean labels and in a negative direction for data with label errors, compared to the simple edge sum. This indicates that the label noisiness score calculated with max-cut better separates label errors from the clean data. In this case, TNR95 increases from 0.628 to 0.647 by applying the max-cut.

### D.3 Label error detection

In this section, we provide the exact performance values for Figure 5, including AUROC results. We report label error detection performances under various noise levels in Table 10, and provide performances according to the number of data in Table 11. The tables confirm that our relation graph approach achieves the best label error detection performances in all three metrics, regardless of the noise ratio and the number of data. In Tables 12 to 14, we provide performance values of all baselines considered in Table 1.

### D.4 OOD detection

In Tables 15 and 16, we provide OOD detection results on individual datasets mentioned in Section 4.2: Places, SUN, iNaturalist, and Textures. We report the OOD detection performance of MAE-Large in Table 15 and the performance of ResNet-50 in Table 16. The tables demonstrate that our approach achieves the best OOD detection performance on three out of four datasets considered, while achieving the best overall performance. Furthermore, our method shows the best performance on all three metrics with both models, which highlights its effectiveness in detecting OOD data.

## E  Additional qualitative results

### E.1 Relation map

In Figure 13, we present additional relation maps on ImageNet. We draw the relation maps with MAE-Large and ResNet-50. Note, MAE-Large utilizes masked auto-encoding pre-training whereas ResNet-50 is trained from scratch. From the figure, we observe that the models exhibit similar distributions of positive and negative relations for each data point. However, the ResNet model tends to have an overall larger variance of relation, indicating that the pre-training process reduces the relation variance and is helpful in forming relationships between data.

The visualization tool helps us to comprehend the distribution of complementary and conflicting relations associated with a data point, aiding dataset analysis and debugging. For example, in Figure 13, the relation map of the Ram sample exhibits a combination of positive and negative relations. Specifically, it exhibits positive relations with other Ram class samples while showing negative relations with samples from the Big Horn class, which are visually challenging to distinguish.

This observation suggests the need for multi-labeling or refinement of the label space definition. In this way, we can intuitively identify problems in the dataset by using the visualization tool.

## E.2 Qualitative results

We present problematic data samples detected by our algorithm. Specifically, Figure 14 shows ImageNet samples with label errors and their most conflicting data samples having negative relation values. Figure 15 shows the outlier data detected by our algorithm on the ImageNet validation set, indicating the existence of inappropriate data for evaluation. We also present results on SST2 [55], a binary text sentiment classification dataset, in Tables 17 and 18.

Table 10: Label error detection performance on a range of label error ratios (MAE-Large, ImageNet).

| Method \ Ratio | AUROC | | | | AP | | | | TNR95 | | | |
|---|---|---|---|---|---|---|---|---|---|---|---|---|
| | 4% | 8% | 12% | 15% | 4% | 8% | 12% | 15% | 4% | 8% | 12% | 15% |
| Entropy | 0.790 | 0.796 | 0.801 | 0.803 | 0.153 | 0.246 | 0.334 | 0.389 | 0.115 | 0.137 | 0.162 | 0.181 |
| Least-conf. | 0.829 | 0.836 | 0.843 | 0.847 | 0.176 | 0.282 | 0.380 | 0.443 | 0.223 | 0.270 | 0.327 | 0.368 |
| CWE | 0.845 | 0.845 | 0.845 | 0.844 | 0.301 | 0.397 | 0.471 | 0.506 | 0.135 | 0.160 | 0.188 | 0.212 |
| Loss | 0.864 | 0.864 | 0.865 | 0.865 | 0.328 | 0.424 | 0.497 | 0.530 | 0.229 | 0.276 | 0.334 | 0.374 |
| TracIn | 0.889 | 0.888 | 0.887 | 0.885 | 0.303 | 0.415 | 0.498 | 0.536 | 0.503 | 0.521 | 0.556 | 0.582 |
| Margin | 0.876 | 0.875 | 0.876 | 0.876 | 0.403 | 0.484 | 0.544 | 0.568 | 0.346 | 0.392 | 0.457 | 0.505 |
| Relation (Ours) | **0.917** | **0.914** | **0.910** | **0.904** | **0.437** | **0.526** | **0.590** | **0.611** | **0.695** | **0.695** | **0.687** | **0.683** |

Table 11: Label error detection performance according to the number of data (MAE-Large, ImageNet, 8% label noise).

| Method \ #data | AUROC | | | | | AP | | | | | TNR95 | | | | |
|---|---|---|---|---|---|---|---|---|---|---|---|---|---|---|---|
| | 12k | 25k | 100k | 400k | 1.2M | 12k | 25k | 100k | 400k | 1.2M | 12k | 25k | 100k | 400k | 1.2M |
| Entropy | 0.806 | 0.801 | 0.797 | 0.796 | 0.796 | 0.250 | 0.248 | 0.248 | 0.246 | 0.246 | 0.161 | 0.161 | 0.144 | 0.137 | 0.137 |
| Least-conf. | 0.844 | 0.841 | 0.836 | 0.837 | 0.836 | 0.289 | 0.287 | 0.284 | 0.283 | 0.282 | 0.255 | 0.269 | 0.275 | 0.277 | 0.27 |
| CWE | 0.852 | 0.850 | 0.845 | 0.845 | 0.845 | 0.390 | 0.394 | 0.392 | 0.394 | 0.397 | 0.169 | 0.184 | 0.177 | 0.163 | 0.16 |
| Loss | 0.869 | 0.868 | 0.863 | 0.864 | 0.864 | 0.415 | 0.421 | 0.418 | 0.421 | 0.424 | 0.260 | 0.298 | 0.282 | 0.284 | 0.276 |
| TracIn | 0.891 | 0.890 | 0.886 | 0.887 | 0.888 | 0.408 | 0.415 | 0.409 | 0.412 | 0.415 | 0.554 | 0.541 | 0.523 | 0.524 | 0.521 |
| Margin | 0.878 | 0.878 | 0.874 | 0.875 | 0.875 | 0.481 | 0.486 | 0.477 | 0.481 | 0.484 | 0.376 | 0.424 | 0.412 | 0.398 | 0.392 |
| Relation (Ours) | **0.910** | **0.912** | **0.912** | **0.914** | **0.914** | **0.502** | **0.515** | **0.518** | **0.523** | **0.526** | **0.679** | **0.692** | **0.692** | **0.696** | **0.695** |

Table 12: Label error detection performance on various scales of MAE (ImageNet, 8% noise).

| Scale | Metric | Entropy | Least-conf. | CWE | Loss | TracIn | Margin | Relation |
|-------|--------|---------|-------------|-----|------|--------|--------|----------|
| Base | AP | 0.231 | 0.280 | 0.373 | 0.412 | 0.393 | 0.477 | **0.514** |
| | TNR95 | 0.120 | 0.217 | 0.132 | 0.223 | 0.488 | 0.342 | **0.672** |
| Large | AP | 0.246 | 0.282 | 0.397 | 0.424 | 0.415 | 0.484 | **0.526** |
| | TNR95 | 0.137 | 0.270 | 0.160 | 0.276 | 0.521 | 0.392 | **0.695** |

Table 13: Label error detection performance on speech (ESC-50) and language (SST2, MNLI) datasets with 10% label noise. For ESC-50, we use the AST model [10], and for SST2 and MNLI, we use the RoBERTa-Base model [30].

| Dataset | Metric | Entropy | Least-conf. | CWE | Loss | TracIn | Margin | Relation |
|---------|--------|---------|-------------|-----|------|--------|--------|----------|
| ESC-50 | AP | 0.715 | 0.737 | 0.720 | 0.737 | 0.739 | 0.737 | **0.779** |
| | TNR95 | 0.784 | 0.783 | 0.790 | 0.783 | 0.789 | 0.793 | **0.847** |
| SST2 | AP | 0.227 | 0.227 | 0.861 | 0.861 | 0.854 | 0.861 | **0.881** |
| | TNR95 | 0.175 | 0.175 | 0.850 | 0.850 | 0.837 | 0.850 | **0.870** |
| MNLI | AP | 0.199 | 0.197 | 0.753 | **0.764** | 0.724 | 0.754 | 0.758 |
| | TNR95 | 0.103 | 0.104 | 0.494 | 0.497 | 0.510 | 0.514 | **0.660** |

Table 14: Label error detection AP on ImageNet validation set. All the model scales are Large.

| Model | Entropy | Least-conf. | CWE | Loss | TracIn | Margin | Relation |
|-------|---------|-------------|-----|------|--------|--------|----------|
| MAE | 0.558 | 0.609 | 0.688 | 0.703 | 0.695 | 0.708 | **0.733** |
| BEIT | 0.614 | 0.644 | 0.707 | 0.719 | 0.718 | 0.718 | **0.737** |
| ConvNeXt | 0.587 | 0.625 | 0.696 | 0.710 | 0.700 | 0.713 | **0.735** |
| ConvNeXt-22k | 0.617 | 0.642 | 0.707 | 0.722 | 0.719 | 0.724 | **0.744** |

Table 15: OOD detection performance on ImageNet with MAE-Large.

| Method \ OOD dataset | AUROC | | | | | AP | | | | | TNR95 | | | | |
|---|---|---|---|---|---|---|---|---|---|---|---|---|---|---|---|
| | ALL | Places | SUN | iNat. | Textures | ALL | Places | SUN | iNat. | Textures | ALL | Places | SUN | iNat. | Textures |
| MSP | 0.857 | 0.835 | 0.833 | 0.907 | 0.853 | 0.818 | 0.543 | 0.567 | 0.699 | 0.514 | 0.428 | 0.386 | 0.315 | 0.651 | 0.376 |
| Max Logit | 0.824 | 0.787 | 0.790 | 0.881 | 0.850 | 0.808 | 0.531 | 0.557 | 0.700 | 0.570 | 0.138 | 0.106 | 0.088 | 0.320 | 0.216 |
| Mahalanobis | 0.875 | 0.819 | 0.841 | 0.948 | 0.904 | 0.815 | 0.471 | 0.489 | 0.722 | 0.532 | 0.521 | 0.407 | 0.474 | **0.847** | 0.599 |
| Energy | 0.776 | 0.728 | 0.738 | 0.829 | 0.837 | 0.757 | 0.446 | 0.467 | 0.592 | 0.553 | 0.074 | 0.060 | 0.050 | 0.127 | 0.154 |
| ReAct | 0.896 | 0.862 | 0.872 | 0.944 | 0.909 | 0.857 | 0.586 | 0.611 | 0.746 | 0.634 | 0.550 | 0.449 | 0.463 | 0.793 | 0.547 |
| KL-Matching | 0.877 | 0.848 | 0.857 | 0.928 | 0.874 | 0.818 | 0.500 | 0.526 | 0.708 | 0.455 | 0.548 | 0.473 | 0.490 | 0.738 | 0.511 |
| Iterative sampling | 0.444 | 0.377 | 0.409 | 0.457 | 0.600 | 0.382 | 0.127 | 0.134 | 0.145 | 0.173 | 0.073 | 0.051 | 0.061 | 0.125 | 0.131 |
| Local outlier factor | 0.556 | 0.492 | 0.521 | 0.594 | 0.665 | 0.431 | 0.151 | 0.160 | 0.183 | 0.173 | 0.156 | 0.113 | 0.136 | 0.283 | 0.240 |
| KNN | 0.901 | 0.861 | 0.884 | 0.946 | **0.922** | 0.854 | 0.550 | 0.593 | 0.736 | **0.648** | 0.590 | 0.487 | 0.560 | 0.808 | 0.626 |
| Relation (Ours) | **0.911** | **0.883** | **0.894** | **0.951** | **0.921** | **0.874** | **0.618** | **0.653** | **0.782** | 0.642 | **0.636** | **0.547** | **0.587** | 0.810 | **0.641** |

Table 16: OOD detection performance on ImageNet with ResNet-50.

| Method \ OOD dataset | AUROC | | | | | AP | | | | | TNR95 | | | | |
|---|---|---|---|---|---|---|---|---|---|---|---|---|---|---|---|
| | ALL | Places | SUN | iNat. | Textures | ALL | Places | SUN | iNat. | Textures | ALL | Places | SUN | iNat. | Textures |
| MSP | 0.847 | 0.829 | 0.836 | 0.896 | 0.814 | 0.782 | 0.469 | 0.501 | 0.641 | 0.369 | 0.496 | 0.471 | 0.461 | 0.634 | 0.352 |
| Max Logit | 0.844 | 0.827 | 0.833 | 0.894 | 0.807 | 0.767 | 0.445 | 0.468 | 0.605 | 0.331 | 0.482 | 0.463 | 0.448 | 0.622 | 0.340 |
| Mahalanobis | 0.693 | 0.644 | 0.628 | 0.735 | 0.823 | 0.567 | 0.228 | 0.209 | 0.270 | 0.375 | 0.265 | 0.194 | 0.229 | 0.436 | 0.371 |
| Energy | 0.836 | 0.821 | 0.825 | 0.883 | 0.798 | 0.721 | 0.398 | 0.407 | 0.503 | 0.269 | 0.481 | **0.463** | 0.448 | 0.622 | 0.340 |
| ReAct | 0.624 | 0.610 | 0.623 | 0.626 | 0.650 | 0.446 | 0.185 | 0.189 | 0.188 | 0.128 | 0.361 | 0.302 | 0.370 | 0.429 | 0.371 |
| KL-Matching | 0.838 | 0.811 | 0.822 | 0.894 | 0.817 | 0.784 | 0.439 | 0.463 | 0.693 | 0.404 | 0.307 | 0.268 | 0.224 | 0.477 | 0.291 |
| Iterative sampling | 0.624 | 0.530 | 0.578 | 0.659 | 0.809 | 0.516 | 0.167 | 0.190 | 0.229 | 0.380 | 0.150 | 0.112 | 0.134 | 0.223 | 0.269 |
| Local outlier factor | 0.648 | 0.556 | 0.602 | 0.686 | 0.824 | 0.533 | 0.175 | 0.199 | 0.244 | 0.397 | 0.186 | 0.143 | 0.167 | 0.272 | 0.315 |
| KNN | 0.822 | 0.743 | 0.783 | 0.884 | **0.922** | 0.764 | 0.335 | 0.384 | 0.546 | **0.774** | 0.380 | 0.295 | 0.372 | 0.587 | **0.496** |
| Relation (Ours) | **0.870** | **0.830** | **0.853** | **0.922** | 0.879 | **0.818** | **0.493** | **0.543** | **0.708** | 0.513 | **0.515** | 0.429 | **0.498** | **0.691** | 0.465 |

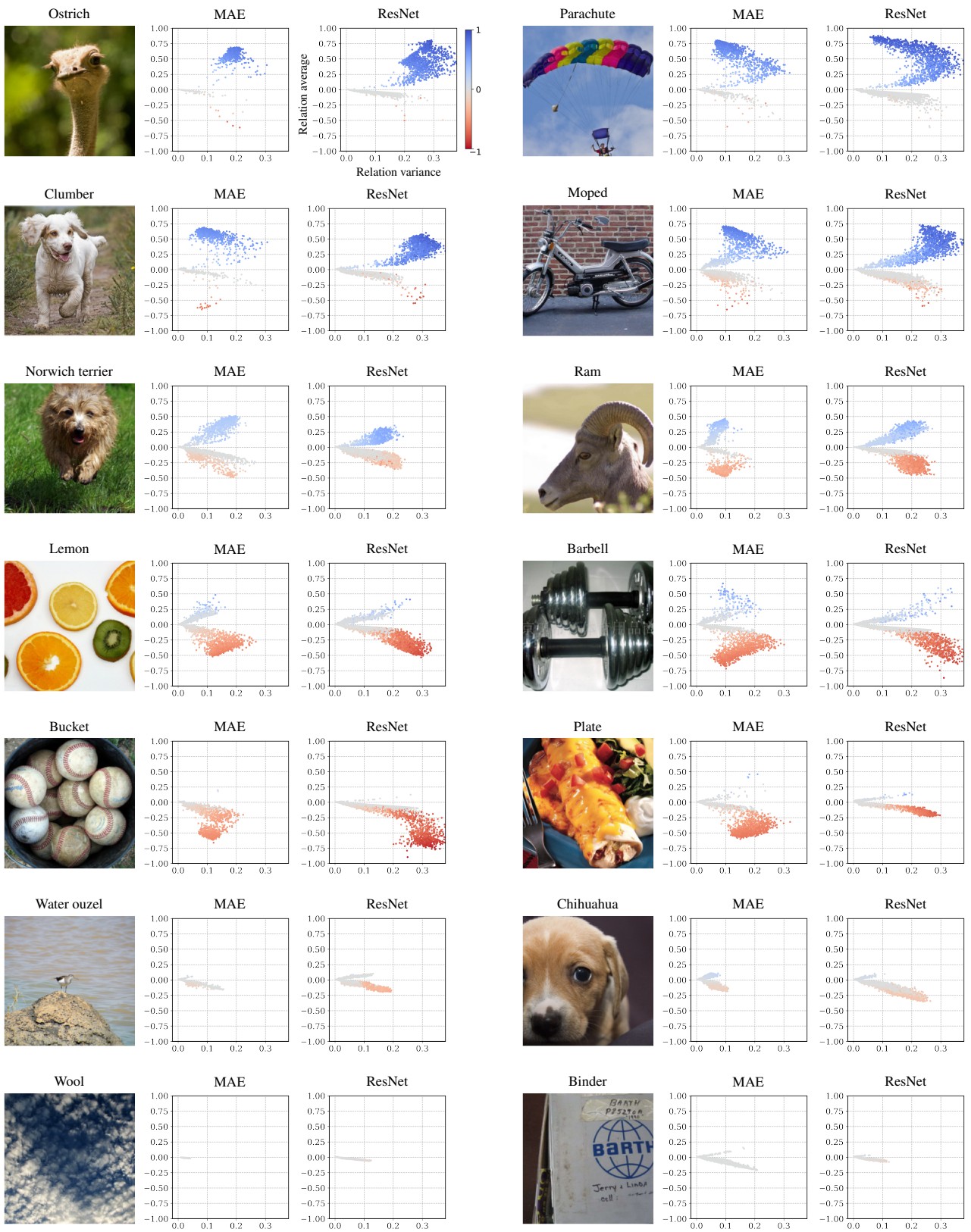

Figure 13: Data relation maps on ImageNet with MAE-Large and ResNet-50. We denote the assigned label above each image. The color represents the relation value at the last checkpoint. The x-axis is the standard deviation and the y-axis is the mean value of the relation values throughout training.

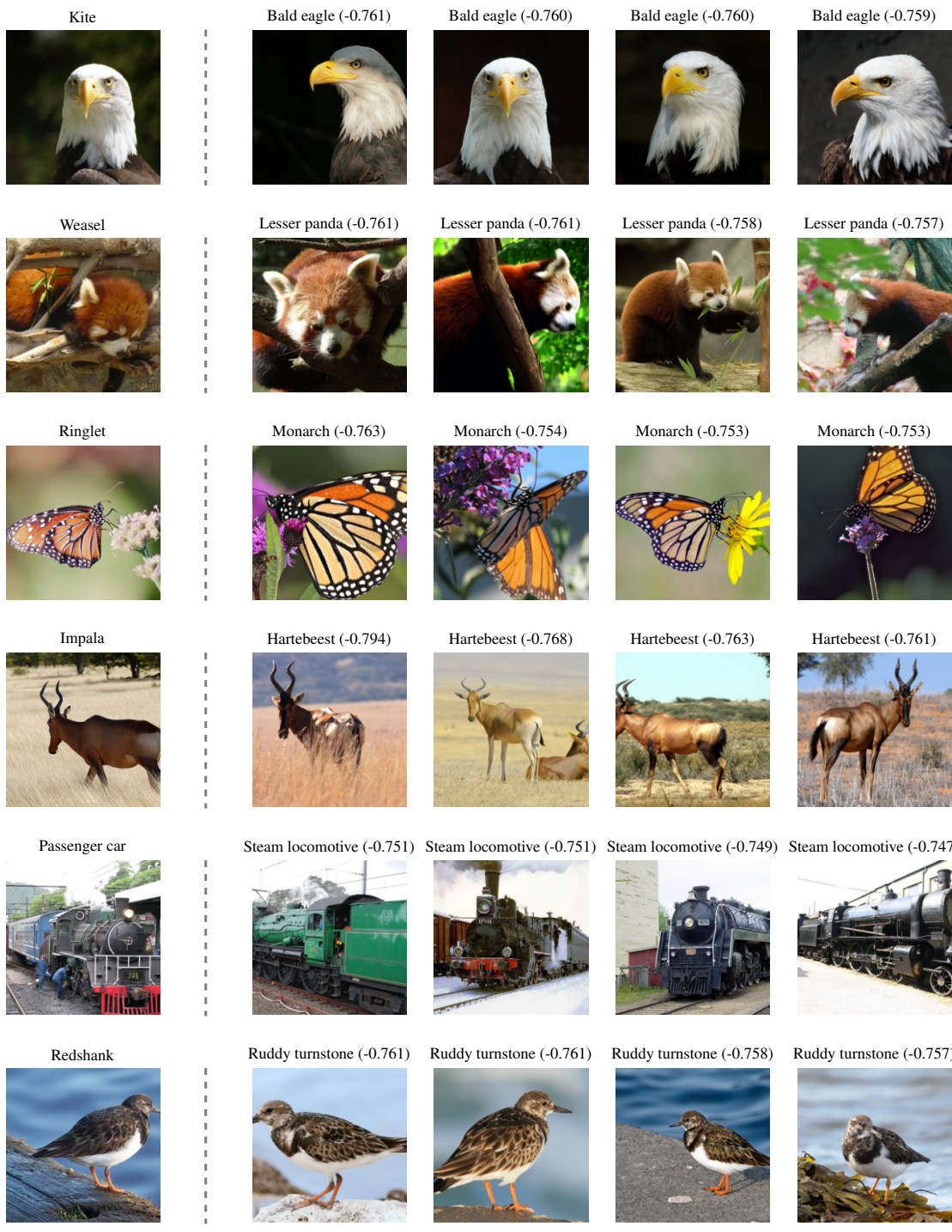

Figure 14: The first column shows the data samples detected by our label error detection algorithm using MAE-Large on ImageNet. We present the samples with the most conflicting relation next to the detected samples. We denote the assigned label and the corresponding relation value above the image.

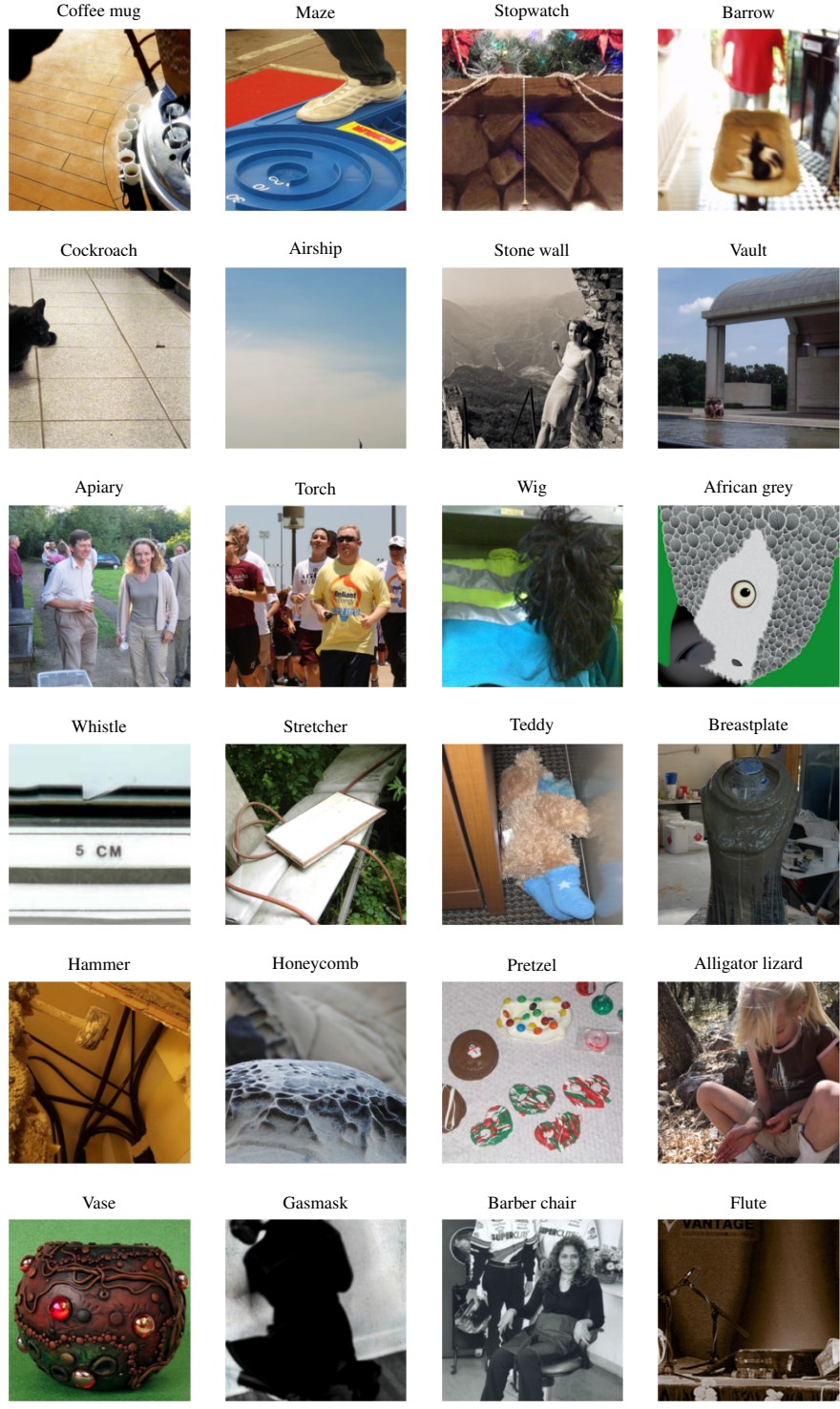

Figure 15: Data samples with the highest outlier score by our method on the ImageNet validation set. We denote the assigned label above the image.

Table 17: Text samples with label errors detected by our algorithm on the SST2 dataset. Below the text with label error, we present two text samples with conflicting relations and denote the corresponding relation value in parenthesis.

| Text | Label |
|------|-------|
| a damn fine and a truly distinctive and a deeply pertinent film | Negative |
|     - a breathtakingly assured and stylish work | Positive (-0.980) |
|     - a winning and wildly fascinating work | Positive (-0.978) |
| fails to have a heart, mind or humor of its own | Positive |
|     - failing to find a spark of its own | Negative (-0.970) |
|     - this movie's lack of ideas | Negative (-0.958) |
| a ploddingly melodramatic structure | Positive |
|     - plodding action sequences | Negative (-0.959) |
|     - plodding picture | Negative (-0.957) |
| is somewhat problematic | Positive |
|     - the more problematic aspects | Negative (-0.945) |
|     - the problematic script | Negative (-0.930) |
| a bittersweet contemporary comedy | Negative |
|     - bittersweet film | Positive (-0.940) |
|     - of bittersweet camaraderie and history | Positive (-0.921) |

Table 18: Outlier text samples detected by our algorithm on the SST2 dataset.

| Text | Label |
|------|-------|
| the battle | Negative |
| give a backbone to the company | Positive |
| leather pants | Positive |
| the israeli/palestinian conflict as | Negative |
| the story relevant in the first place | Positive |
| from a television monitor | Positive |
| loud, bang-the-drum | Positive |
| a movie instead of an endless trailer | Negative |
| a doctor's office, emergency room, hospital bed or insurance company office | Positive |
| this is more appetizing than a side dish of asparagus | Negative |

