# OpenReview forum: "Neural Relation Graph: A Unified Framework for Identifying Label Noise and Outlier Data"
_NeurIPS.cc/2023/Conference — NeurIPS 2023 poster_

### Official Review · Reviewer_Yk3p · 2023-07-05

**Soundness:** 3 good
**Presentation:** 3 good
**Contribution:** 3 good
**Rating:** 6
**Confidence:** 3

**Summary:**

This paper proposes a novel approach called the Neural Relation Graph framework for identifying label noise and outlier data in large-scale datasets with real-world distributions. The approach utilizes a relational structure of data in the feature-embedded space to detect label errors and outlier data, and introduces a visualization tool for interactive data diagnosis. The authors conduct extensive experiments on various tasks and demonstrate that their approach achieves state-of-the-art detection performance and is effective in debugging real-world datasets. The contributions of this paper include a unified approach for diagnosing and cleaning large-scale datasets, a data relation function and graph algorithms for detecting label errors and outlier data, and a visualization tool for interactive data diagnosis.

**Strengths:**

Originality:
The Neural Relation Graph framework proposed in this paper is a novel approach for identifying label noise and outlier data in large-scale datasets. The authors utilize a relational structure of data in the feature-embedded space to detect label errors and outlier data, which is a unique and innovative approach. The paper also introduces a visualization tool for interactive data diagnosis, which is a novel contribution to the field. Overall, the paper is highly original and presents a new perspective on diagnosing and cleaning large-scale datasets.

Quality:
The paper is of high quality, with a well-designed methodology and extensive experiments conducted on various tasks. The authors provide detailed descriptions of the proposed approach and the experiments conducted, which makes it easy to understand and replicate the results. The paper also includes a thorough evaluation of the proposed approach, comparing it to existing methods and demonstrating its effectiveness in detecting label errors and outlier data. The quality of the paper is further enhanced by the use of clear and concise language, making it easy to follow and understand.

Clarity:
The paper is well-written and easy to understand, with clear descriptions of the proposed approach and the experiments conducted. The authors provide detailed explanations of the technical terms used, making it accessible to a wide range of readers. The paper also includes visual aids, such as figures and tables, which help to illustrate the concepts presented. Overall, the clarity of the paper is excellent, making it easy to follow and understand.

Significance:
The paper is highly significant, as it presents a novel approach for diagnosing and cleaning large-scale datasets with real-world distributions. The proposed approach utilizes a relational structure of data in the feature-embedded space to detect label errors and outlier data, which is a unique and innovative approach. The paper also introduces a visualization tool for interactive data diagnosis, which is a valuable contribution to the field. The results of the experiments conducted demonstrate the effectiveness of the proposed approach, making it a significant contribution to the field of machine learning.

**Weaknesses:**

One potential weakness of the paper is that the authors do not provide a detailed analysis of the limitations of their approach. While the proposed approach achieves state-of-the-art detection performance on various tasks, it is unclear how it would perform on datasets with different characteristics or in different domains. The authors could address this weakness by conducting experiments on a wider range of datasets and providing a more detailed analysis of the limitations of their approach.

Another weakness of the paper is that the authors do not provide a detailed discussion of the computational complexity of their approach. While the paper mentions that the proposed algorithms are scalable, it is unclear how they would perform on very large datasets or in real-time applications. The authors could address this weakness by providing a more detailed analysis of the computational complexity of their approach and discussing potential strategies for improving its scalability.

Finally, the paper could benefit from a more detailed discussion of the practical implications of the proposed approach. While the paper demonstrates the effectiveness of the approach in detecting label errors and outlier data, it is unclear how it could be applied in real-world scenarios. The authors could address this weakness by discussing potential use cases for the proposed approach and providing guidance on how it could be integrated into existing machine learning pipelines.

Overall, the paper presents a novel and innovative approach for diagnosing and cleaning large-scale datasets, but could benefit from a more detailed analysis of its limitations, computational complexity, and practical implications.

**Questions:**

Can you provide a more detailed analysis of the limitations of your approach? While the proposed approach achieves state-of-the-art detection performance on various tasks, it is unclear how it would perform on datasets with different characteristics or in different domains.

Can you provide a more detailed discussion of the computational complexity of your approach? While the paper mentions that the proposed algorithms are scalable, it is unclear how they would perform on very large datasets or in real-time applications.

Can you discuss potential use cases for the proposed approach and provide guidance on how it could be integrated into existing machine learning pipelines? While the paper demonstrates the effectiveness of the approach in detecting label errors and outlier data, it is unclear how it could be applied in real-world scenarios.

Can you provide more details on the visualization tool introduced in the paper? While the tool is mentioned briefly, it would be helpful to have a more detailed description of its functionality and how it can be used to diagnose data.

Can you provide more details on the datasets used in the experiments? While the paper mentions that experiments were conducted on various tasks, it would be helpful to have more information on the characteristics of the datasets and how they were selected.

Can you provide more details on the hyperparameters used in the experiments? While the paper mentions that hyperparameters were tuned using cross-validation, it would be helpful to have more information on the specific values used and how they were selected.

Can you provide more details on the implementation of the proposed algorithms? While the paper mentions that the algorithms were implemented using PyTorch, it would be helpful to have more information on the specific implementation details and any potential optimizations that were made.

Can you discuss potential future directions for this research? While the paper presents a novel and innovative approach, it would be helpful to have a discussion on potential future directions for this research and how it could be extended or improved upon.

**Limitations:**

The paper does not explicitly address the potential negative societal impact of the proposed approach. While the focus of the paper is on diagnosing and cleaning large-scale datasets, it is possible that the approach could be used for other purposes, such as identifying individuals or groups based on their data. This could potentially lead to privacy concerns or other negative societal impacts.

However, it should be noted that the paper does not provide any evidence that the proposed approach has been used for such purposes, and the authors do not make any claims about the potential negative societal impact of their work. Additionally, the paper does not explicitly address the limitations of the proposed approach, which could potentially lead to unintended consequences if the approach is used in real-world scenarios.

Overall, while the paper does not explicitly address the potential negative societal impact of the proposed approach, it should be noted that the authors do not make any claims about the potential negative impact of their work, and the focus of the paper is on diagnosing and cleaning large-scale datasets.

---

> ### Author Rebuttal · Authors · 2023-08-07
>
> We thank you for your valuable efforts and time in providing insightful feedback. We would like to address the questions below.
>
> **Limitations of the proposed approach (with different domains)**
> - Thank you for your feedback. In Table 1-b from Sec 4.1.2, we demonstrate that our method consistently outperforms baselines across **various data types, including image, speech, and text** domains. Meanwhile, we acknowledge some limitations of our work.
> - Firstly, our current approach is confined to classification tasks. Expanding the application of our method to a more diverse set of tasks, such as generative modeling or segmentation, is an important future direction. Furthermore, we observed from Figure 5 (b) that the performance gap between our approach and the baselines narrows as the number of data points in the relation graph decreases. While our method demonstrates superior performance even with 12k data from ImageNet, it is essential to address this limitation of decreased performance when dealing with a small number of data points. We will include these discussions in the main text.
>
> **Discussion on computational complexity**
> - We would like to highlight our theoretical complexity analysis in L159-166. We demonstrate that when utilizing parallel computing, the complexity of our algorithm is $O(k^2)$ for $k\ll n$, with $n$ representing the total number of data points. In Appendix A.3, we examine the time taken by the algorithm to process a large-scale dataset (1.2M ImageNet) in a real-world computing environment. The results show that our algorithm takes only a few minutes and comprises only 5~20% of the execution time for feature extraction, demonstrating its efficiency in handling large-scale datasets.
>
> **Potential use cases**
> - Thank you for your suggestion. Our method can be applied to various application scenarios, including data annotation, model evaluation, and robust inference. For instance, our problematic data detection algorithms can aid human annotators during the data annotation process. In addition, as shown in Appendix Figure 13 and Table 19, our algorithm allows for the effective detection and removal of outlier data from the evaluation set, leading to a more accurate model evaluation. Furthermore, our algorithm can determine whether a data point is out-of-distribution during inference, enabling a more reliable inference system. We will incorporate these discussions in the revised version.
>
> **Details on the visualization tool**
> - The visualization tool described in Section 3.5 helps us to comprehend the distribution of complementary and conflicting relations associated with a data point, aiding dataset analysis and debugging. For example, in Appendix Figure 11, the relation map of the Ram sample exhibits a combination of positive and negative relations. Specifically, it exhibits positive relations with other Ram class samples while showing negative relations with samples from the Big Horn class, which are visually challenging to distinguish. This observation suggests the need for multi-labeling or refinement of the label space definition. In this way, we can intuitively identify problems in the dataset by using the visualization tool. We will elaborate on these explanations in the revised version.
>
> **Details on the datasets**
> - Thank you for your feedback. The brief description and references for the datasets are provided in lines 239-243. We will update the dataset descriptions in a table format, including the number of data points and classes. These datasets were selected from the baseline methods, focusing on those with large scales. For example, ImageNet was used in the TracIn paper, and MNLI was used in Dataset Cartography. We will update this detailed information in the revision.
>
> **Hyperparameter details**
> - Thank you for your comment. In Appendix C, Table 5, we provide a summary of the hyperparameters used in our method. As noted, we used a **fixed set of hyperparameters** for each task (label error detection, OOD/outlier detection) across various datasets and models during the evaluation. We performed hyperparameter tuning on ImageNet, guided by the hyperparameter sensitivity analysis presented in Figure 8 and Table 6. The results indicate that our method does not require specific tuning for each model or dataset, enabling efficient adaptation in real-world applications.
>
> **Implementation details**
> - Our implementation is based on PyTorch, and we conducted each experiment on a single RTX3090-Ti GPU. To optimize computation efficiency, we utilized a caching technique to store the initial label noisiness scores, as described in lines 140-143. Moreover, in Appendix C.1, lines 599-601, we provided an implementation detail for handling small noisy values. We will release the detailed source code for reproducibility.
>
> **Potential future direction**
> - Thank you for your feedback. We briefly described the future direction of our approach in Appendix B, lines 568-574. We believe expanding our approach beyond classification to encompass tasks like generative modeling or segmentation is an important future direction. Furthermore, integrating our method into a real-world data annotation process is one of the future directions. We will elaborate on these points in the main text.
>
> **Potential negative social impact**
> - Our research primarily focuses on tackling technical challenges related to label errors and outliers, and does not directly relate to sensitive social issues. Nevertheless, we acknowledge the possibility that the algorithm could be utilized to detect specific group data, potentially leading to social implications. During the revision process, we will emphasize in the main text that the primary objective of our paper is strictly confined to addressing technical issues and does not advocate any discriminatory usage of the algorithm.
>
> Thank you once again for the valuable feedback. If you have any remained questions, please let us know.

---

> > ### Comment · Area_Chair_diXz · 2023-08-18
> > **Awaiting Your Feedback on Authors' Rebuttal**
> >
> > Dear Reviewer Yk3p,
> >
> > Thank you for your hard work. The Author-Reviewer discussion ends on August 21. The authors and I are eager to learn whether their responses have adequately addressed your concerns. You are encouraged to directly reply to the authors' rebuttal.
> >
> > Please note that this is a public thread. If you prefer to reply to me individually, please use the internal discussion thread.
> >
> > Kind Regards,
> >
> > AC

---

> ### Comment · Area_Chair_diXz · 2023-08-16
> **Discussion**
>
> Dear Reviewer Yk3p,
>
> The authors have provided their response. Can you please get in touch with them to assess if their response meets your criteria? If not, could you highlight any remaining concerns? Thank you very much for your help.
>
> Best Regards,
>
> AC

---

### Official Review · Reviewer_Cygf · 2023-07-06

**Soundness:** 3 good
**Presentation:** 4 excellent
**Contribution:** 3 good
**Rating:** 7
**Confidence:** 3

**Summary:**

The authors identified the issue of how existing label errors in the training data and the OOD in the test set can affect the model training and evaluation and further proposed a novel approach utilizing the learned feature embeddings and label information to compute the relations between data instances. The relations represent how similar the two data instances are in terms of their feature embeddings and also assigned labels. This is further used to construct a relational graph structure. Based on the graph structure, they introduce a min-cut algorithm based on the label noisiness score to identify a subset of label errors. They further visualized the derived graph structure for the purpose of interactive data error diagnosis. Extensive experiments are conducted on multiple data types (images, audio, texts), and all show superior performance over the baseline methods. Through comparison, authors showed their relational data structure provides complementary information not captured by the unary scoring methods. Generally, this paper is well-written, the ideas are pretty clear, the methods are novel, and results are solid.

**Strengths:**

A substantive assessment of the paper's strengths, touching on each of the following dimensions: originality, quality, clarity, and significance. We encourage reviewers to be broad in their definitions of originality and significance. For example, originality may arise from a new definition or problem formulation, creative combinations of existing ideas, application to a new domain, or removing limitations from prior results. You can incorporate Markdown and Latex into your review. See /faq.

1.	This paper is well-written, and the authors explain the ideas clearly. The method is novel, and experiments and results are solid.
2.	The example provided in Figure 1 explained well the limitations of using the unary scoring method when identifying the label error and the outlier data.
3.	The design of the data relation function in Section 3.1 looks simple but still effective.
4.	In section 3.2, the authors pointed out that simply aggregating all edges of a node can yield suboptimal results, which makes much sense, and they further proposed using the min-cut method as a walkaround. This looks interesting.
5.	The experiments are conducted using various data types, including images, speeches, and texts. This also covers multi-class tasks and binary tasks. All experiments show the effectiveness of their methods, which is solid.


**Weaknesses:**


1.	In Figure 1, the difference between the Coil (label error) and the Envelope (outlier) is not very clear. More elaborations will be helpful.
2.	Constructing the relational graph is a fully connected graph, the time complexity would be O(N^2), and the scalability to a large dataset is a concern.
3.	In section 3.2, pp3 line 115, ‘a lower (label noisiness) score indicates a higher likelihood of label error’. A lower label noisiness indicates a higher error possibility, this is a bit confusing and not very intuitive.


**Questions:**


1.	In Figure 1, What makes the Coil example a label error, while the Envelope example an outlier? Some elaboration on this would be helpful.
2.	In Section 3.2, pp3 line114, authors set $r(i,i)=0$, this is not very straightforward and the reason is not clear to me. More elaboration on this would be helpful.
3.	In section 3.2, pp4 line 127, authors minimize the sum of the edges between two groups. One question comes naturally - what about the sums of edges within each group?
4.	In terms of different datasets used in this work, several data types were involved (images, speeches, texts), it would be helpful to see what the embedding dimensions are for each dataset.


**Limitations:**

Minor edit suggestion: pp8 line309, ‘3.6%p’ should be ‘3.6%’.

---

> ### Author Rebuttal · Authors · 2023-08-07
>
> We thank you for your valuable efforts and time in providing insightful feedback on our work. We would like to address questions from the reviewer below.
>
> **Difference between the Coil (label error) and the Envelope (outlier)**
> - In the case where a data point is associated with a different ground truth label, it is considered a label error, and when it is not possible to assign a suitable label, the data point is referred to as an outlier. For instance, in the case of the "Coil" example, it corresponds to the "park bench" label in ImageNet (incorrectly labeled as "coil" in the actual ImageNet dataset), and for the "envelope" example, a suitable ImageNet label cannot be found for the given data. We will clarify these explanations in the revision.
>
> **The complexity of constructing the relational graph**
> - Our method employs a simple cosine feature similarity for relation values, which involves a simple matrix multiplication of $n$ pre-computed features. This computation can be efficiently carried out on GPUs. In practice, extracting features from 1.2M ImageNet data points using an MAE-Large model takes 15x longer time than the execution time of Algorithm 1 (Appendix A.3, Table 3). Specifically, the computation of Algorithm 1 with full ImageNet can be accomplished within a few minutes by using a single GPU. Furthermore, as demonstrated in the complexity analysis in lines 159-166, we can improve complexity to O(k^2), for $k \ll n$, leveraging partitioning and parallel computing techniques.
>
> **A lower label noisiness indicates a higher error possibility, this is a bit confusing and not very intuitive**
> - Thank you for the pointer. We will modify the expression to be more intuitive.
>
> **More elaboration on the assumption r(i,i)=0**
> - Our approach only considers relations between different data points, which means the value $r(i, i)$ is not taken into account in the algorithm. However, if the formula is expressed considering this value, the notation becomes complicated, and to enhance the clarity of the expressions, we have chosen to set this value to 0. We will make this more clear in the revised version.
>
> **About the sums of edges within each group**
> - That's a great point. The sum of all edges in the graph is equal to the sum of edges between groups plus the sum of edges within each group. Since the total sum of edges for a specific dataset is fixed, minimizing the sum of edges between groups is equivalent to maximizing the sum of edges within each group. This new perspective is highly intriguing, and to aid the reader's understanding, we will incorporate this information into the main text.
>
> **Embedding dimensions of each dataset**
> - Thank you for the point. The embedding dimensions for each case are as follows:
> | Image (MAE-Large) | speech (Ast) | text (RoBERTa-Base) |
> |:-:|:-:|:-:|
> | 1024 | 768 | 768 |
>
> - We will include the information above in the revision.
>
> **Minor edit suggestion: pp8 line309, ‘3.6%p’ should be ‘3.6%’**
> - Thank you for the pointer. We will fix it in the revision.
>
> Thank you once again for the valuable feedback. If you have any remained questions, please let us know.

---

> > ### Comment · Reviewer_Cygf · 2023-08-14
> >
> > Thanks for the reply! I have read the responses.

---

> > > ### Author Response · Authors · 2023-08-15
> > >
> > > Dear reviewer, thank you for confirmation!

---

### Official Review · Reviewer_6KY3 · 2023-07-06

**Soundness:** 3 good
**Presentation:** 3 good
**Contribution:** 2 fair
**Rating:** 6
**Confidence:** 3

**Summary:**

This paper proposes a new method using graph structure for detecting label errors and outlier data. Briefly speaking, the algorithm utilizes data feature embeddings to generate relation graph and using the new defined data relation function, its algorithm can capture mislabeled and outlier data from a global prospective. A large number of experimental results show that the performance of this method is greatly improved compared with the existing detection methods.

**Strengths:**

(1) This paper presents a novel relation graph-based approach to achieve better utility and considers error label and outlier detection in a global way.
(2) Extensive experiments on datasets from different domains show the improvement of the new method is promising.
(3) Comprehensive ablation study is performed, which helps to understand the proposed model better.
(4) The overall expression of the article is clearer and easier to understand, and related works are adequately cited.


**Weaknesses:**

(1) Certain statements in this paper lack justification. For example, the method is based on building graph on feature space, but there is a little discussion about feature spaces and how to generate it.
(2) Lack of sufficient theoretical analysis (e.g. convergence but not only empirical analysis) , the proof of proposition and analysis of relation function are relatively simple.
(3) Lack of explanation of choosing parameter in baseline method, like whether there is a finetune on the choice of K in KNN method.


**Questions:**


(1)	Line 159 in Complexity analysis, the paper demonstrates the complexity can be reduced to O(nk). But in the fourth last row in algo 1 each point in partition seems also need to calculate n relation score instead of k. Can you deeper explain the improvement in complexity？ And there is no empirical results of the acceleration.
(2)	The paper mentions it’s the first using data relation graph on the feature space. The “first” points to “relation graph” or “feature space”? Are there any other methods using graph to detect error data and need to be compared with?


**Limitations:**

(1)	Although extensive experiments on real-world datasets corroborate the effectiveness of the proposed method, there is lack of theoretical analysis of the effectiveness of the method.
(2)	Some content lacks detailed explanations.

---

> ### Author Rebuttal · Authors · 2023-08-07
>
> We thank you for your valuable efforts and time in providing insightful feedback on our work. We would like to address questions from the reviewer below.
>
> **Discussion about feature spaces**
> - Thank you for the pointer. We would like to mention that specific information regarding the feature space can be found in Appendix C.1, lines 589-590. We used neural networks trained on a noisy training set as feature extractors. Specifically, we followed the references of TracIn, MAE, and RoBERTa, using the input vector to the classification layer of neural networks as features. We will incorporate this information into the main text in the revision.
>
> **About theoretical analysis**
> - Thank you for your response. In Proposition 1, we theoretically prove the convergence of our proposed algorithm, which is also empirically demonstrated in Appendix A.2. We believe that the simplicity and intuitiveness of our proof offer an advantage, as they make the concepts more accessible to a broad readership, ensuring a clear understanding of our work.
>
> **Hyperparameter of baselines**
> - Thank you for your question. We tuned the hyperparameters of the baseline methods by adhering to the instructions provided in the respective papers, as mentioned in Section 4.3, lines 314-315. Specifically, for the KNN method, we followed the formula presented in the paper, which specifies that $k=n_{\text{class}} \times \alpha$ (where alpha represents the ratio of training data used) for the tuning. We will elaborate on these details in the revision. Additionally, we will make the source code, including the baseline methods, publicly available to further support reproducibility.
>
> **Questions on complexity (+empirical comparison)**
> - Figure 5 (b) demonstrates that our method maintains superior detection performance even with a small number of data in the relation graph, e.g., 1% of the ImageNet training set. Based on the observation, the complexity analysis suggests that when given a dataset with $n$ samples, dividing it into $n/k$ partitions of size $k$ and running the algorithm on each partition improves the complexity. In this case, the complexity for each partition is $O(k^2)$, and when applied sequentially to $n/k$ partitions, the complexity becomes $O(k^2 * n / k) = O(nk)$. Following the suggestion, we empirically validate the complexity of our Algorithm on ImageNet with 1 RTX-3090Ti GPU:
> |# data | 0.1M | 1M (accelerated) | &nbsp;&nbsp;1M |
> |:--|:--:|:--:|:--:|
> | Time (s) | 3 | 31 | 424 |
> | AP | 0.518 | 0.522 | 0.526 |
>
> - In the table above, we examine the computation time and label error detection performance of our algorithm on ImageNet with MAE-Large. Specifically, the column labeled 1M (accelerated) represents the results of our algorithm's accelerated version, achieved through partitioning of size 0.1M. Considering that the performance of the best baseline is 0.484, our algorithm demonstrates efficient acceleration while maintaining the best performance. We will incorporate this information in the revised version.
>
> **The “first” points to “relation graph” or “feature space”?**
> - We appreciate your feedback and would like to clarify the point mentioned in lines 83-85. The term "first" was used to indicate the order of the contents in the following sections rather than claiming to be the initial proposer of the concept. Our main contribution lies in presenting a unified framework, novel relation graph structure, and effective algorithms, which differ from the baseline methods, primarily relying on a single score of each feature. While there are other methods that also utilize relationships between data, such as the KNN method, it's important to note that this approach is limited to OOD issues and does not leverage a global relational structure. As a result, our proposed approach demonstrates higher performance, as presented in Figure 7, page 9.
>
> Thank you once again for the valuable feedback. If you have any remained questions, please let us know.

---

> ### Comment · Area_Chair_diXz · 2023-08-16
> **Discussion**
>
> Dear Reviewer 6KY3
>
> The authors have provided their response. It would be greatly appreciated if you could communicate with the authors to confirm whether their response addresses your concerns, or to specify any remaining issues.
>
> Many Thanks,
>
> AC

---

> > ### Comment · Reviewer_6KY3 · 2023-08-17
> > **My concerns have been addressed**
> >
> > Thanks for your response. I'll keep my score.

---

### Official Review · Reviewer_nM5G · 2023-07-26

**Soundness:** 3 good
**Presentation:** 3 good
**Contribution:** 2 fair
**Rating:** 5
**Confidence:** 4

**Summary:**

The paper under review outlines a novel method utilizing graph structure to identify label errors and outlier data. It proposes an algorithm that makes use of data feature embeddings to produce relation graphs. By incorporating a newly defined data relation function, the algorithm can globally capture mislabeled and outlier data. The experiments conducted exhibit a marked improvement in performance compared to extant detection methods.

**Strengths:**

+ Innovation: The paper introduces a novel method, which is an important contribution to the field. This new approach potentially provides a fresh perspective and further insights into the problem at hand.

+ Clarity and Comprehensiveness: The paper is well-structured and clearly written, making it easy for readers to understand the content. The authors have adequately cited related works, showing a thorough understanding of the existing literature and situating their work appropriately within that context.

+ Important Problem: The paper tackles an important problem, making its potential impact highly relevant and timely. This problem is pertinent to many real-world scenarios, amplifying the value of the proposed solution.

**Weaknesses:**

+ Unclear Advantage: Despite the introduction of a new method, the paper lacks clear explanation of the motivation and advantages of the chosen approach compared to existing methods. The authors proposed to use the feature similarity ( 'semantic similarity between data points') to help learn a noise-robust classifier, which exploits $P(X)$ to help learn $P(Y|X)$. It shares the same underlying physiology with semi-supervised based methods (e.g., DividMix, ICLR20) or self-supervised based methods (e.g., UNICON, CVPR22).

+ Lack of Explicit Assumptions: The authors do not clarify the assumptions under which the proposed method is expected to perform well. This lack of clarity could impede understanding and application of the method in practical scenarios.

+ Unknown Motivation for Kernel Usage: The motivation for using a kernel in the proposed method is not explained. Without this, it's hard to understand the reason behind the choice of a kernel and how it contributes to the method's effectiveness.

+ Absence of Baseline Comparisons: The paper does not compare the proposed method with significant baselines in the field of learning with noisy labels (e.g., DividMix, ICLR20; ELR, NeurIPS20; CausalNL, NeurIPS21; C2D, WACV23; UNICON, CVPR22). Such comparisons are crucial for evaluating the performance of the new method and understanding its standing relative to existing techniques.

**Questions:**

+ Could you elaborate on the unique advantages of your proposed method over existing semi-supervised and self-supervised techniques, especially considering the shared approach of using feature similarity to help learn a noise-robust classifier? What distinguishes your method from others that also exploit $P(X)$ to help learn $P(Y|X)$?

+ Could you specify the assumptions under which your method is expected to perform well?

+ Could you provide the reasoning behind the choice of using a kernel in your method? How does the kernel contribute to the effectiveness of the method and why was this specific kernel chosen over potential alternatives?

+ To convince others about the method's practical significance, could you include a comparison of your method with DividMix (ICLR20) and  UNICON (CVPR22) in the field of learning with noisy labels?

**Limitations:**

I did not come across a sufficient discussion of the limitations of the proposed method. I would highly recommend the authors include a section on potential limitations in their revision.

---

> ### Author Rebuttal · Authors · 2023-08-05
>
> We thank you for your valuable efforts and time in providing insightful feedback. We would like to address questions below.
>
> **Advantage over the semi-/self-supervised techniques**
> - Thank you for the pointer. We would like to clarify that the goal of our paper is to identify the problematic data not only during training but also in more general situations such as evaluation and inference. Our method has some advantages over the mentioned semi-/self-supervised techniques:
> 1. Our approach has **various other applications**, including data annotation, evaluation set debugging, and robust inference (lines 17-20), whereas the semi-/self-supervised methods focus on training with noisy labels. For instance, as shown in Table 1-c, our method can effectively detect label issues in the validation set, enabling a more accurate evaluation system. Furthermore, as shown in Appendix Figure 13, our method can identify whether a data point is an outlier, which helps build a reliable inference system.
> 2. Our method does **not rely on specific training techniques**, making it applicable to more general data types and models. For instance, DivideMix and UNICON require mixup training, which is not commonly used in text domains, e.g., GPT-3 [1]. Our method assumes that the model is given, allowing it to be effortlessly applied to multiple domains using different training techniques, as demonstrated in Table 1-b.
> [1] Brown et al., "Language Models are Few-Shot Learners", 2020
> 3. Our method does **not demand additional training costs**, making it easier to scale to large-scale models and datasets. The semi-/self-supervised methods like DivideMix and UNICON require training two networks and numerous augmented training samples for SSL. Our approach, on the other hand, only needs a single trained network and identifies problematic data without additional training.
>
> - In the revision, we will incorporate the aforementioned points into the related work section.
>
> **Comparison with DivideMix and UNICON**
> - Thank you for the valuable suggestion. While the mentioned methods aim to train image classifiers with noisy labels, our approach is focused on the task of identifying and debugging problematic data in different scenarios and domains. Technically, we focus on determining whether a data point $x_i$ or its label $y_i$ is anomalous, whereas the mentioned methods aim to directly infer a clean label $y_i$, making a direct comparison less straightforward. For instance, due to the difference, our approach is applicable to outlier detection as well, whereas those baselines cannot be applied.
> - We recognize that DivideMix and UNICON proposed partitioning algorithms for clean/noisy labels using GMM and Uniform Clean Sampling. However, it's worth noting that their approaches rely on sample-level cross-entropy or JSD loss, while our method explicitly models the relation between data. We reproduced the baseline algorithms using publicly available GitHub code and compared them with our relation graph approach:
> |dataset \ method|DivideMix|UNICON|Relation (ours)|
> |:-|:-:|:-:|:-:|
> |ImageNet|0.424|0.447|0.526|
> |ESC-50|0.737|0.739|0.779|
> |MNLI|0.754|0.762|0.766|
>
> - The table above compares the label error detection AP over various datasets in different domains (Table 1-a,b of our paper). Our approach demonstrates better performance compared to DivideMix and UNICON, confirming the advantages of our method across various datasets in label error detection. We acknowledge the importance of these related works and we will incorporate the results into the revision.
>
> **Assumptions under which the method is expected to perform well**
> - Thank you for the pointer. In our approach, we assume the availability of a model trained on the noisy training dataset (line 89). As illustrated in Figure 6-left, when the model is insufficiently trained (e.g., trained for 10 epochs), the label noise detection performance tends to decrease, as the model does not capture meaningful semantic relationships between the data. It is worth noting that the model training requirement is a common assumption for other baseline techniques as well.
> - Furthermore, as evident from Figure 5, the performance of our algorithm improves as the number of data points in the relation graph increases. This analysis suggests that our method's performance can be effectively enhanced when handling massive data in real-world applications. We will ensure to present these discussions more clearly in our revision.
>
> **Motivation and reasoning behind the kernel used**
> - To evaluate our relation graph framework, we adopted the most commonly used and computationally efficient kernel, i.e., feature cosine similarity. Notably, our framework demonstrates strong performance with other kernels, e.g., RBF kernel (Sec 4.3. Table 2), indicating that the effectiveness of our approach primarily stems from the graph structure and algorithm rather than the specific kernel design.
> - Another notable advantage of the kernel described in Section 3.4 is interpretability. In lines 200-211, we establish a relationship between the proposed kernel and the influence function. Additionally, in Appendix A.4, we provide a formal explanation demonstrating that the proposed kernel can exhibit better robustness to outliers compared to the influence function.
>
> **Potential limitations**
> - Thank you for your feedback. Due to space limitations, we included the limitations and future work in Appendix B, line 568. Currently, we have evaluated our method within the scope of classification. Applying it to a broader range of tasks, such as generative modeling or segmentation, is an important future endeavor. Additionally, establishing a more rigorous theoretical foundation for our approach is also an important future direction. We will supplement these points and move them to the main text in the revision.
>
> Thank you once again for the valuable feedback. If you have any remained questions, please let us know.

---

> > ### Comment · Reviewer_nM5G · 2023-08-17
> > **My concerns have been addressed**
> >
> > Dear authors,
> >
> > Thank you very much for your clear response. I have no further concerns. I have updated my rating.
> >
> > Kind regards,
> >
> > Reviewer nM5G

---

> ### Comment · Area_Chair_diXz · 2023-08-16
> **Discussion**
>
> Dear Reviewer nM5G,
>
> The authors have provided their response. Can you please get in touch with them to assess if their response meets your criteria? If not, could you highlight any remaining concerns? Thank you very much for your help.
>
> Best Regards,
>
> AC

---

### Author Rebuttal · Authors · 2023-08-07

Dear reviewers,

Thank you for your valuable effort and time in providing helpful feedback on our work. We sincerely appreciate your encouraging comments, including:

- The paper introduces a novel method, provides a fresh perspective into the problem, and is highly original. (Reviewer nM5G, 6KY3, Yk3p)
- The paper tackles an important problem, making its potential impact highly relevant and timely (Reviewer nM5G)
- Extensive and solid experiments show the improvement of the new method is promising. (Reviewer 6KY3, Cygf)
- The clarity of the paper is excellent (Reviewer Yk3p)

We have carefully considered all the points raised and provided rebuttals accordingly. If you have any further questions, please let us know. We will reflect all discussions in the revision and release the code for reproducibility.

Best regards,
The authors

---

### Comment · Area_Chair_diXz · 2023-08-12
**Discussion period**

Dear Reviewers,

I would like to express my sincere gratitude for your thorough examination of this paper. Now that the authors have provided their rebuttal, I kindly ask you to evaluate whether their response sufficiently addresses the concerns you have raised.
Should you require any additional information or have further questions, please feel free to request clarification directly from the authors. Your insights and contributions to this process are greatly appreciated!

Best regards,

AC

---

### Decision · Program_Chairs · 2023-09-21

**Decision:**

Accept (poster)

**Comment:**

This paper introduces a unified method that leverages graph structures to detect label errors and outliers. The core idea is to build a relation graph by using training data and identify label errors or outliers using a min-cut algorithm based on a noisiness score.

During the rebuttal, reviewers acknowledge that 1). A novel approach is proposed for identifying label noise and outlier; 2). contributes to an important problem; 3). The written quality overall is high.

Reviewers have also voiced some concerns about the clarity regarding the method's limitations, its unique advantages, the omission some baselines, and possible issues related to computational complexity. After the response by the authors, the raised concerns are properly addressed. One reviewer increased score. All reviewers' scores are now positive.

AC recommends acceptance of this paper. For the camera-ready version, the authors should integrate the reviewers' feedback and implement the suggested changes. For the convenience of readers, it would also be great to include a brief survey of learning with noisy labels, e.g., covering model-free methods and transition-matrix-based methods.